# INSM1 governs a neuronal progenitor state that drives glioblastoma in a human stem cell model

Patrick A. DeSouza [1,2], Matthew Ishahak [3,16], Xuan Qu [2,4,5,16], Colin McCornack [1,2,16], Devi Annamalai[1], Diane D. Mao[1], Rajanikanth Vangipurapu[1], Yiwei Fu[1], Alexandre T. Vessoni [6,7,8], Ryan T. Cleary[1,2], Rowland H. Han [1], Punn Augsornworawat [3,9,10], Timothy Woodiwiss[1,11], Darby Agovino[6,7], Braxton Sizemore [1,2], Jessica Kline[1,2], Maryam Borhani-Haghighi[1], Hao Chen[2,12], Sangami Pugazenthi [1], Hiroko Yano [1,2,13,14], Ting Wang [2,4,5,15], Luis F. Z. Batista[6,7], Jeffrey R. Millman [3,9] ✉ & Albert H. Kim [1,2,12,13,14] ✉

Glioblastoma is a lethal brain cancer marked by functional plasticity driven by tumor cell-intrinsic mutations and their interplay with developmental programs. To investigate how canonical glioblastoma mutations promote functional plasticity, we have developed an isogenic human neural stem cell (NSC) model of glioblastoma by sequential addition of *TERT* promoter, *TP53*, and *PDGFRA* point mutations. TP53 loss-of-function increases *TERT* expression during serial mutagenesis, but only triple mutant NSCs reliably form lethal brain tumors in vivo that recapitulate glioblastoma. Tumor cell evolution triggers stress-related metabolic changes and transitions toward a neuronal progenitor network driven by transcription factor INSM1. INSM1 is highly expressed in human glioblastoma tumors and, during cortical development, in intermediate progenitor cells, which give rise to neurons. Remarkably, *INSM1* knockdown in triple mutant NSCs and primary glioblastoma cells disrupts oncogenic gene expression and function and inhibits the in vivo tumorigenicity of triple mutant NSCs, highlighting the functional importance of an intermediate progenitor cell-like cell state in glioblastoma pathogenesis.

Glioblastoma (GBM) is an aggressive, fatal brain tumor that displays marked inter- and intra-tumoral heterogeneity[1–5]. Tumor cells evade maximum therapeutic interventions with almost inevitable recurrence, but the precise molecular and cellular mechanisms of adaptive persistence remain elusive. GBM is likely initiated in multipotent progenitor cell niches, including the subventricular zone, a site where driver mutations have been observed[6,7]. However, despite extensive cataloging of GBM driver mutations, clinical trials targeting driver genes have not yet significantly improved or stratified patient survival[8]. There is a lack of common post-therapy mutations during tumor

relapse, necessitating a deeper understanding of the epigenetic contributions to tumor heterogeneity and disease progression[9].

Bulk molecular profiling initially uncovered three dominant gene expression signatures that classified GBM transcriptomes in patients, called Proneural, Classical, and Mesenchymal[3]. However, when measuring genome-wide RNA levels in single cells became possible, each subtype was detected intratumorally in varying proportions within individual patients with the dominant subpopulation driving bulk classification[4,10]. A spatially weighted analysis of tumor cell states in GBM revealed that cells resembling neural and oligodendrocyte

progenitors correlated with developmental features as well as the Proneural (PRO) signature[11].

The association between PRO gene expression and the infiltrative tumor edge has motivated closer examination of hijacked developmental mechanisms during invasion. Recent work has demonstrated that whole-brain invasion and colonization of GBM is fueled by cells resembling neuron- and neural progenitor-like tumor states which receive synaptic input from surrounding neurons[12]. In association with these states, neuronal activity induced complex calcium signals in GBM cells, which stimulated invasion, indicating how GBM dissemination and neurodevelopmental phenotypes are closely linked[13]. Mutations in *TP53*, *PDGFRA*, and *CDK4* are tightly associated with the PRO signature and developmental features of invasion[10,11]. However, the epigenetic drivers that link these canonical GBM mutations to the transcriptional and cellular plasticity that enable malignant developmental cell states remain largely unknown.

Here, we develop an isogenic human embryonic stem cell (hESC)-derived model of PRO GBM, harboring mutations in the *TERT* promoter, *TP53* binding domain, and *PDGFRA* kinase domain. Following differentiation and validation of wildtype (WT) and mutant engineered neural stem cells (eNSCs), we intracranially implant these cells in athymic nude mice and observe robust and lethal brain tumor formation by only the triple mutant PRO eNSCs. Using this isogenic GBM model for transcriptomic and chromatin accessibility analyses at single cell resolution, we uncover an evolutionary trajectory of tumorigenic cells from stress-related metabolic changes to neuronal progenitor-like expression networks during disease progression and identify critical epigenetic drivers of this process. Our findings directly link genetic mutations associated with GBM to oncogenic gene expression patterns, developmental states, and chromatin regulators, which are required for tumor cell evolution and function.

## Results

### Developing a human embryonic stem cell-derived model of the GBM Proneural subtype

To investigate the epigenetic regulators of GBM pathogenesis in a defined genetic context, we performed a mutant-specific analysis of tumor cell heterogeneity using a human neural stem cell (NSC) model. Activating mutations in the *TERT* promoter (*TERT*p) are among the most common, early selective drivers of GBM formation[9]. Because the mutation occurring 124 bp upstream of the *TERT* transcription start site (C228T) occurs in up to 90% of GBMs, we utilized CRISPR/Cas9-mediated genome editing to produce heterozygous *TERT*p C228T mutant human embryonic stem cells (ESCs) in two independent genetic clones (Methods). We next added a heterozygous *TP53* dominant negative (DN), gain-of-function mutation (G473A → R248Q) to the *TERT*p mutants to produce two genetic clones of double mutant *TERT*p + *TP53* hESCs (Fig. 1A, B, and Supplementary Fig. 1A). Matched wildtype (WT) control cells were exposed to CRISPR/Cas9 reagents but remained non-mutagenized. We generated NSCs from WT and mutant ESCs by adapting a planar differentiation protocol[14]. This resulted in decreased expression of the ESC marker OCT4 and increased expression of the NSC marker PAX6 (Fig. 1C, and Supplementary Fig. 1B, C) in two independent clone sets. Importantly, there were no significant differences in expression of cell state markers between WT and mutant genotypes, indicating a negligible effect of genetic background on NSC differentiation (Supplementary Fig. 1C–E). Following lentiviral transduction of kinase-activated mutant *PDGFRA*[D842V] (vs. control empty vector), we confirmed receptor expression and activity (Fig. 1D, Supplementary Fig. 1F), and used two clones each of the four serial mutant engineered NSC (eNSC) genotypes (WT, *TERT*p, *TERT*p + *TP53*, and triple mutant PRO) for further study.

### PRO eNSCs display increased tumorigenicity and recapitulate GBM pathology

To begin understanding the functional changes driven by stepwise addition of each mutation in eNSCs, we measured in vitro transformation by self-renewal capacity and clonogenic growth on soft agar (Fig. 1E–G, and Supplementary Fig. 1G-I). Two independent genetic clones of PRO eNSCs displayed an increase in stem cell frequency and clonogenic growth compared to the other mutant and WT cells. When injected orthotopically in athymic nude mice, two independent genetic clones of PRO eNSCs reliably formed large brain tumors detectable by MRI and caused animal death with a combined median survival of 109 days (Fig. 1H, I, and Supplementary Fig. 2A–C). Smaller lesions were observed in a fraction of mice injected with *TERT*p + *TP53* eNSCs with the majority displaying no signs of malignant growth. Histopathology of PRO eNSC tumors revealed diffuse hemispheric invasion (Fig. 1J, and Supplementary Fig. 2D). PRO eNSCs formed large intraparenchymal tumors, demonstrating characteristic features associated with glioblastoma, including vesicular, spindle-like glial cells with pleomorphism (Fig. 1K, and Supplementary Fig. 2E). Hypercellularity was observed with mitotic atypia and sparse necrosis with vascular proliferation. Salient malignant features also included occasional leptomeningeal dissemination.

### Epigenetic activation of the mutant TERT promoter increases TERT expression and telomerase activity in eNSCs

The *TERT*p mutation increases tumor cell fitness by upregulating *TERT* expression and telomerase activity[15]. We first tested whether the heterozygous *TERT*p C228T mutation increases *TERT* expression and activity in our eNSCs and indeed saw a significant upregulation of both in two independent genetic clones of *TERT*p mutant cells (Fig. 2A, B, and Supplementary Fig. 3A). It was previously shown that the mutant *TERT*p creates a de novo ETS factor motif that can coordinate with a flanking native site to recruit the heterotetrameric GA-binding protein (GABP) transcription factor (TF) complex[16,17] (Fig. 2C). Activation of the mutant promoter leads to stable epigenetic changes, such as alterations in levels of histone tail post-translational modifications (PTMs)[18]. We utilized Cleave Under Nuclease and Release Under Targets (CUT&RUN, Methods) coupled with qRT-PCR to measure levels of activity-dependent histone PTMs at the *TERT*p locus and observed a significant increase of H3K4me3 in *TERT*p mutant eNSCs compared to WT cells (Fig. 2D, and Supplementary Fig. 3B). We also confirmed mutant-specific occupancy of the GABP TF complex at the *TERT*p by chromatin immunoprecipitation (ChIP) coupled with qRT-PCR (Fig. 2E).

Interestingly, we found a consistent increase in *TERT* expression in *TERT*p + *TP53* mutant eNSCs compared to single *TERT*p mutants, which correlated with increased telomerase activity in double mutants (Fig. 2F, G, and Supplementary Fig. 3C). To further investigate this phenomenon, we knocked down both wildtype and mutant *TP53* in our eNSCs using a validated TP53 RNA interference lentivirus[19,20] and observed a significant increase in *TERT* expression in only the single *TERT*p mutants, to levels comparable to *TERT*p + *TP53* double mutants (Supplementary Fig. 3D, E). This suggests that *TP53* loss-of-function rather than gain-of-function mutant activity increases *TERT* expression in the context of a *TERT*p mutation. To understand the relevance of this finding in human GBM, we analyzed GBM bulk RNA and exome data from The Cancer Genome Atlas (TCGA) and found significantly increased *TERT* expression in *TP53* mutant tumors compared to tumors with wildtype *TP53* (Supplementary Fig. 3F).

### Multiplexed transcriptomics in single eNSCs reveal PRO-specific metabolic changes

To understand the evolution of transcriptional changes during serial addition of PRO mutations in our eNSC model, we utilized antibody-based hashtag oligos (HTOs) to uniquely label our four genotypes for

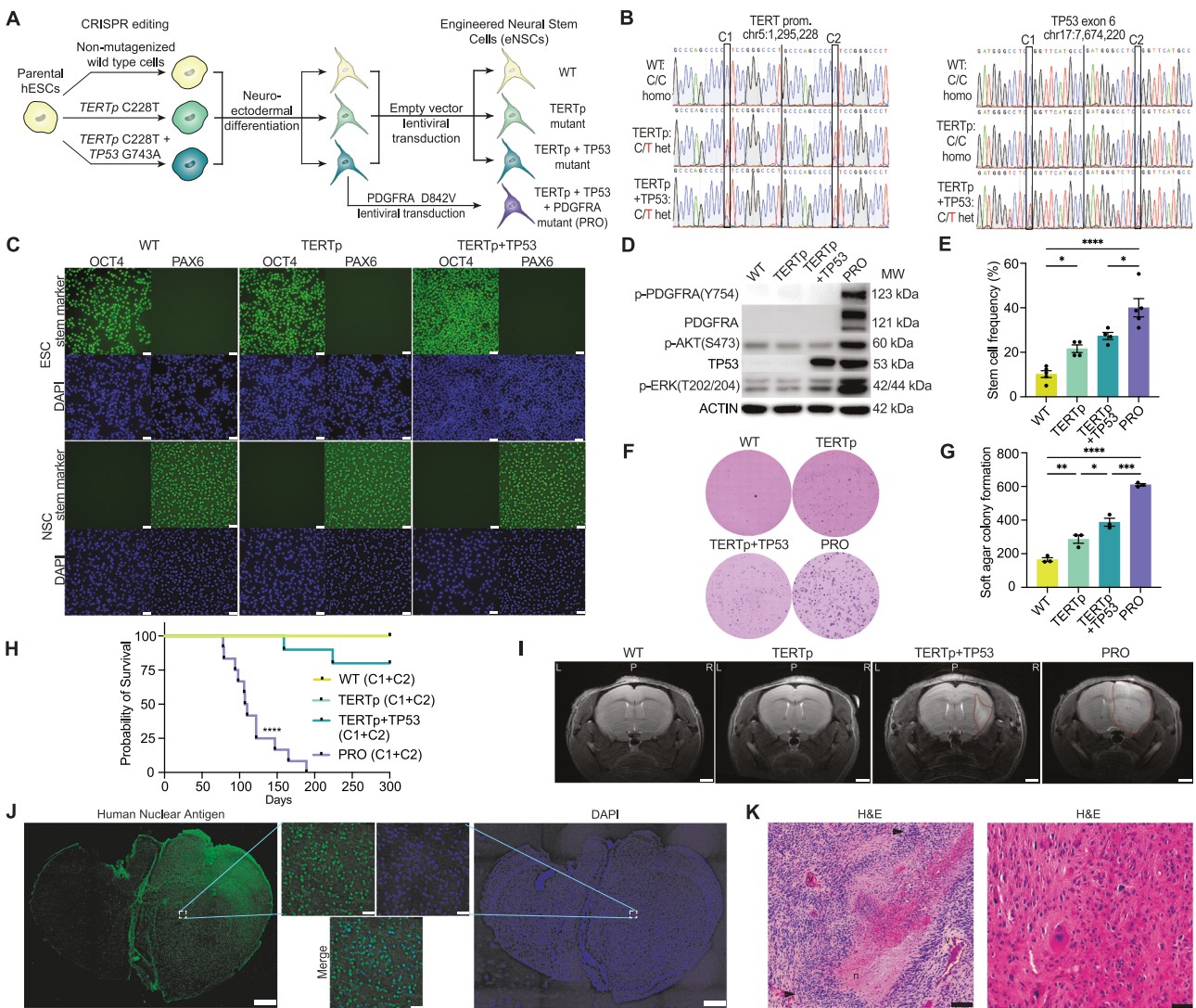

**Fig. 1 | Engineered NSCs form tumors in mice that recapitulate GBM.**
**A** Workflow for generating four serial mutant Proneural eNSC genotypes: Wildtype (WT), *TERT promoter (TERTp) C228T* mutant, *TERTp* mutant + *TP53 G743A* (R248Q) mutant, *TERTp* mutant + *TP53* mutant + *PDGFRA* D842V mutant (triple mutant = PRO). **B** Sanger sequencing validation of CRISPR/Cas9-mediated *TERT*p and *TP53* exon 6 point mutations in 2 independent genetic clones of wildtype, single, and double mutant hESCs. **C** Immunofluorescence for stem cell markers OCT4 and PAX6 (488 nm), as well as DAPI, in WT, *TERT*p, *TERT*p + *TP53* hESC clone set 1 and their differentiation-matched NSCs (scale bar = 50 μM). Shown is a representative image from 3 replicates independently plated for analysis. **D** Immunoblot analyses for mutant PDGFRA expression and activity in one set of four serial mutant eNSC clones. Representative of three replicates. **E** Extreme limiting dilution assay using clone set 1 of the four serial mutant eNSC genotypes. Data represent mean ± SEM, independently plated for analysis (WT and PRO: *n = 4*, *TERT*p and *TERT*p + *TP53*: *n = 3*, ANOVA, *\*P < 0.05, \*\*\*\*P < 0.0001*). **F** Representative soft agar colony formation assay of the four serial mutant eNSC genotypes (clone set 1) and **G** quantification of 3 independent replicates. Data represent mean ± SEM (*n = 3*, ANOVA, *\*P < 0.05, \*\*P < 0.01, \*\*\*P < 0.001, \*\*\*\*P < 0.0001*). **H** Kaplan-Meier survival analysis of four serial mutant eNSCs (two genetic clones, C1 and C2, pooled per genotype) following orthotopic implantation in mice (PRO: *n = 12* animals, other: *n = 10* animals, log-rank, *\*\*\*\*P < 0.0001*). **I** T2-weighted MRI images of mice 100 days post-xenograft with four serial mutant eNSC lines (clone set 1). Shown are representative images from the experimental cohorts. Red outlines indicate pathological lesions in the coronal plane (scale bar = 1 mm; *L* left, *P* posterior, *R* right). **J** Immunofluorescence imaging of representative histological sections generated from PRO eNSC brain tumors (clone 1) stained for Human Nuclear Antigen (488 nm) and DAPI (scale bar = 0.5 mm). Inset images display magnified fields of view and merged images (scale bar = 50 μm). Shown is a representative subject from the experimental animal cohort. **K** Hematoxylin & Eosin (H&E) staining of a representative PRO eNSC brain tumor (clone 1) demonstrating characteristic histopathology of high-grade gliomas. Left: A pink necrotic center (*n*) is surrounded by pseudopalisading tumor cells (arrowheads) and vascular proliferation (v), (left, scale bar = 200 μm). Right: A magnified view of tumor hypercellularity and mitotic atypia (right, scale bar = 50 μm). Shown are representative images from a subject within the experimental animal cohort. Source data are provided as a Source Data file, including exact statistical analyses and *p* values.

multiplexed single cell RNA-sequencing (scRNA-seq) of 11,404 cells (Supplementary Fig. 4A). After the sequencing data was demultiplexed (Supplementary Fig. 4B), we found that principal component analysis (PCA) was unsuccessful in identifying meaningful differences in programs among the different eNSC genotypes, and we thus utilized two alternative and validated approaches to understand the developmental relationships and biological processes in our eNSCs[21,22].

The number of expressed genes per cell, or RNA content, is a robust determinant of trajectory state and direction in scRNA-seq data[21]. We applied this developmental framework to our four eNSCs genotypes using the CytoTRACE algorithm and observed an inferred serial mutation ordering that began with WT cells then decreased in transcriptional diversity toward single *TERT*p, double *TERT*p + *TP53*, and triple mutant PRO eNSCs (Fig. 3A, and Supplementary Fig. 4C).

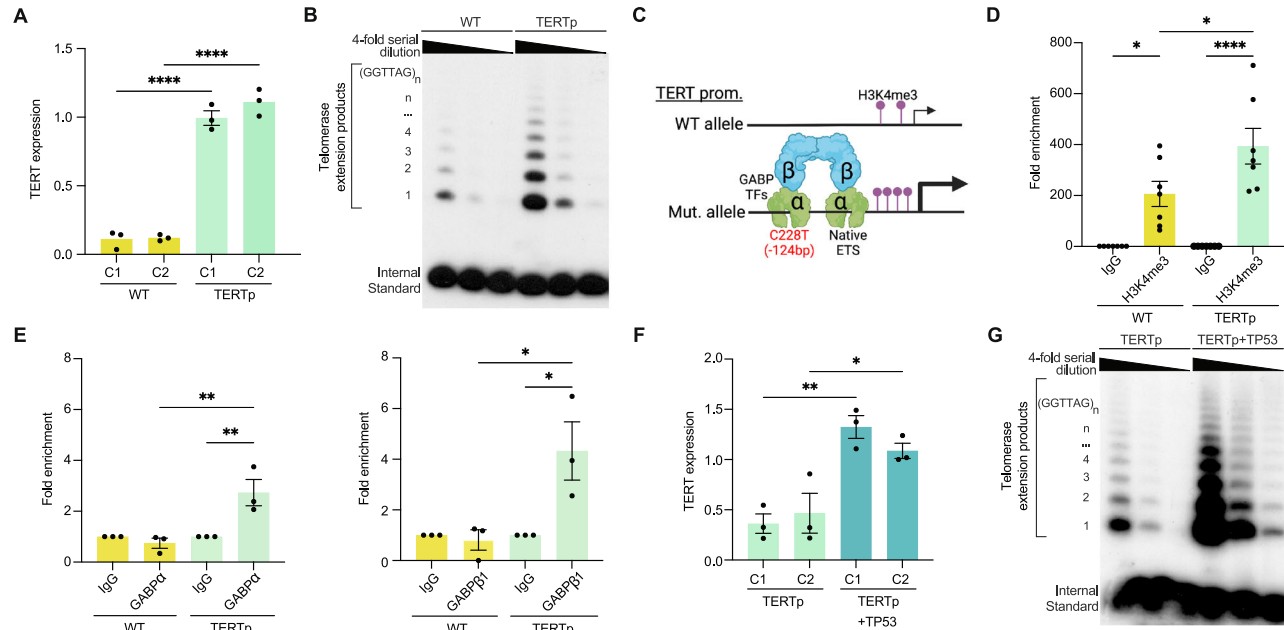

**Fig. 2 | *TP53* loss-of-function increases TERT expression in GBM. A** Quantitative RT-PCR (qRT-PCR) measuring *TERT* mRNA expression normalized to *ACTIN* and *GAPDH* expression in two independent genetic clones of WT and *TERT*p mutant eNSCs. Data represent mean ± SEM in independent replicates (*n = 3*, ANOVA, ****$P < 0.0001$). **B** Telomere Repeat Amplification Protocol (TRAP) assay of WT and *TERT*p mutant eNSCs (clone set 1) measuring endogenous telomerase activity. Each sample is analyzed with two four-fold serial dilutions along with a positive control for DNA amplification (internal standard). The molecular ladder indicates the number of telomeric repeats added to substrate. Data are representative of 3 independent replicates. **C** Schematic of epigenetic dependencies for expression of the monoallelic C228T mutant *TERT* promoter. **D** Cleave Under Targets & Release Under Nuclease coupled with qRT-PCR for analysis of tri-methyl histone 3 lysine 4 (H3K4me3) levels at the *TERT* promoter in WT and *TERT*p eNSCs (clones 1 + 2). Data represent mean ± SEM in independent replicates normalized to IgG (=1) (*n = 5*, ANOVA, *$P < 0.05$, ****$P < 0.0001$). **E** Chromatin Immunoprecipitation coupled with qRT-PCR to measure occupancy of transcription factors GABPα (left) and GABPβ1 (right) in WT and *TERT*p mutant eNSCs (clone set 1). Data represent mean % input ± SEM in independent replicates normalized to IgG (=1) (*n = 3*, ANOVA, *$P < 0.05$, **$P < 0.01$). **F** Quantitative RT-PCR measuring *TERT* mRNA expression normalized to *ACTIN* and *GAPDH* expression in two independent genetic clones of *TERT*p and *TERT*p + *TP53* mutant eNSCs. Data represent mean ± SEM in independent replicates (*n = 3*, ANOVA, *$P < 0.05$, **$P < 0.01$). **G** TRAP assay of *TERT*p and *TERT*p + *TP53* mutant eNSCs (clone set 1). Assay was performed as in (**B**). Data are representative of 3 independent replicates. Source data are provided as a Source Data file, including exact statistical analyses and *p* values.

Functional and motif analyses of genes correlated with PRO eNSCs revealed significant enrichment of pathways related to hypoxia, glycolysis, and neuronal development (Fig. 3B, and Supplementary Fig. 4D). Interestingly, single and double mutant eNSCs largely displayed features of stress-related metabolism, including protein ubiquitination and turnover, as well as global epigenetic signaling, with less prominent features of neuronal development (Supplementary Fig. 4E). We validated the metabolic changes observed in the PRO eNSC analyses in vitro using Seahorse XF analysis and observed a significant increase in glycolysis-dependent extracellular acidification in PRO eNSCs compared with wildtype cells (Fig. 3C). While there were no differences in basal oxygen consumption, PRO eNSCs displayed an elevated spare respiratory capacity, which reflects an increased responsiveness to energy demands under stress (Fig. 3C, and Supplementary Fig. 4F).

**RNA velocity uncovers stable neuronal progenitor trajectories during serial mutagenesis**

We next investigated whether molecular drivers of neuronal specification could be identified in our serial mutation data using a previously published, high-resolution mathematical model of transcriptional processes[22]. RNA velocity, or the time derivative of gene expression state, can be directly estimated by distinguishing spliced and unspliced transcripts as a high-dimensional vector that predicts the future state of individual cells on the timescale of hours[23]. A recently developed computational framework, Dynamo, relies on this concept and further reconstructs continuous vector fields that predict cell fate and underlying regulation of the inferred trajectories[22].

When we applied this algorithm to our scRNA-seq data, a UMAP reduction based on splicing dynamics in eNSCs clearly distinguished the gene expression landscape of mutant eNSCs from WT cells (Fig. 3D). When overlaid with the resultant RNA velocity vector field, we observed a dichotomous separation of transcriptional fate trajectories starting with WT cells and transitioning through serial mutant genotypes. We utilized Monocle3[24] for graph-based pseudotime analysis to validate the biological significance of inferred ordering in the dimension-reduced splicing space (Supplementary Fig. 5A), and observed concordant temporal trajectories terminating in the mutant-specific transcriptional landscape. Using the identified vector field function of the data, we then characterized the topology of the full vector field space by delineating subpopulations in our mutant eNSCs and determining stable, potentially tumorigenic gene expression networks. Following cluster analysis of the splicing-based graph reduction, we characterized the stability of transcription networks within the defined space and identified fixed points that correspond to initiating cell types and more terminal cell types in the inferred trajectory (Fig. 3E, and Supplementary Fig. 5B). We identified several stable gene expression networks with high confidence in the splicing-based transcriptional space that were specific to mutant eNSC subpopulations. These indicate expression networks with high selective potential in the mutant landscape. When we simulated cell fate transitions within this topography using WT cells as the initiating population, we observed a dominance of nodes 16 and 3 as terminal, stable expression patterns for mutant eNSCs (Fig. 3E).

To identify the gene expression changes associated with the tumorigenic PRO eNSCs during serial mutation, we focused on node 16, where a higher representation of PRO cells and higher degree of

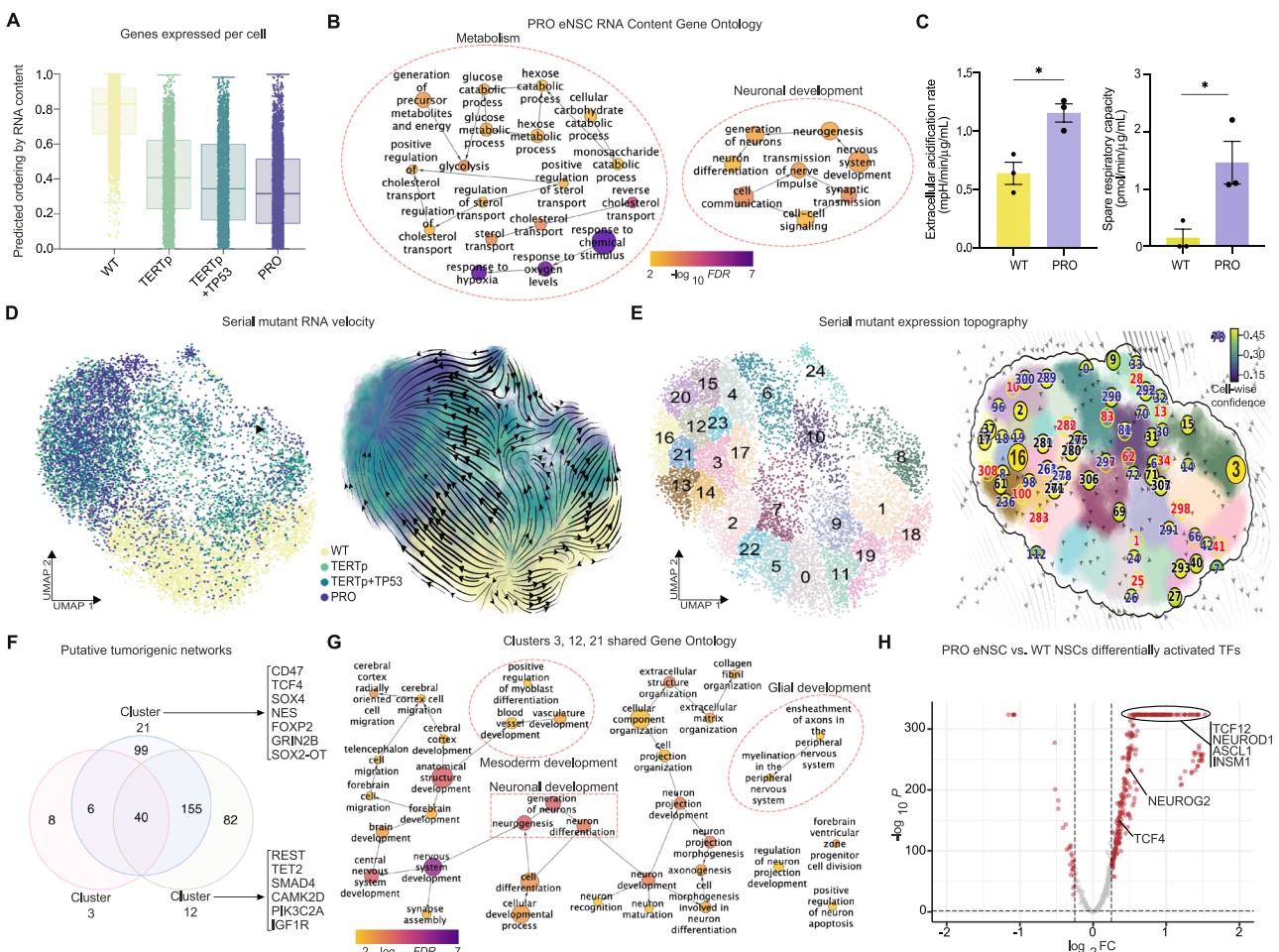

**Fig. 3 | Single cell multiomic analyses of mutant eNSCs in vitro uncovers deregulated metabolism and neuronal fate commitment. A** Inferred ordering of eNSC genotypes by the number of genes expressed in single cells, RNA content, and association with transcriptional covariance across top genes during predicted trajectory (CytoTRACE[21]). Box plots indicate the median (center line), interquartile range (hinges), and 1.5x interquartile range (whiskers). **B** Gene ontology analysis of genes correlated with terminal states predicted by RNA content (n = 405 genes). Circle size represents number of genes within each ontology, and color indicates log-transformed FDR. **C** Seahorse XF analysis in WT and PRO eNSCs of glycolysis-dependent extracellular acidification rate (left) and spare respiratory capacity (right), a measure of metabolic fitness and adaptation to energetic demands. Data represent mean ± SEM in independent replicates (n = 3, ANOVA, **P < 0.01). **D** UMAP reduction of eNSC transcriptomic data by RNA velocity in single cells colored by genotype (left) and overlaid with the RNA velocity vector field (right). **E** Cluster analysis of velocity-based UMAP (left) and expression topography analysis of stable gene networks exhibiting elevated selective potential by dynamical modeling (right). Expression nodes are numbered and represent fixed points within

the landscape. They are colored by cell-wise statistical confidence of identification. Half circles are saddle points within the transcriptomic space while full circles are stable points. Font color indicates absorbing fixed points (black), emitting fixed points (red), and unstable fixed points (blue). Larger nodes represent dominant fixed stable points during fate trajectory simulation using WT eNSCs as the initiating population (Dynamo[22]). **F** Venn diagram of putative tumorigenic clusters falling within the stable absorbing node identified by RNA velocity expression topography analysis. Each circle represents the differentially expressed genes (DEGs) in clusters 3, 12, and 21, and their overlap of shared or unique genes. Shown are representative GBM- or development-related genes identified. **G** Gene ontology analysis of shared DEGs between clusters 3, 12, and 21 (n = 201 genes). Circle size represents number of genes within each ontology, and color indicates log-transformed FDR. **H** Joint analysis of gene expression and chromatin accessibility in single eNSCs for transcription factor motif enrichment and activation in PRO eNSCs compared to WT NSCs. Source data are provided as a Source Data file, including exact statistical analyses and p values.

transcription heterogeneity was identified by cluster analysis (Fig. 3E, and Supplementary Fig. 5B-D). Since node 16 occurred at the borders of clusters 3, 12, and 21, we performed differential gene expression testing and observed shared gene expression networks dominated by clusters 12 and 21 (Fig. 3F). Cluster 21-specific genes were enriched for neurodevelopmental pathways compared to cluster 12 and included *TCF4*, *SOX4*, *FOXP2*, and *GRIN2B* (Supplementary Fig. 5C, D). These genes were of particular interest given their described roles in neuronal specification and GBM invasiveness[25,26]. When we analyzed shared genes between clusters 3, 12, and 21, we found the largest and most significant pathway enrichment of early neuronal development during brain formation, as well as a small representation of plasticity toward glial and vascular development (Fig. 3G). This led us to believe

that while PRO mutations maintain multipotency, there is a selective preference of PRO eNSCs for neuronal cell states and functions related to the developing brain.

We next sought to determine how functional states defined in GBM tumor cells correlate with the gene expression changes observed during in vitro mutagenesis. To address this, we utilized the ontology-based classification scheme developed in Garofano et al.[27] to perform a pathway analysis of our serial mutant eNSCs and scored single cells by enrichment of mitochondrial (MTC), glycolytic/plurimetabolic (GPM), proliferating progenitor (PPR), and neuronal (NEU) functions (Fig. 5SE). We observed decreases in PPR and MTC compartments and the expansion of GPM and NEU oncogenic functions in PRO eNSCs compared to WT cells.

## Multiomic profiling dissects neuronal progenitor TF activation in PRO eNSCs

Recent work has demonstrated that GBM cells hijack neuronal mechanisms for brain invasion[12]. Given the relevant molecular and functional framework of our PRO model, we wondered whether multidimensional epigenetic profiling could provide additional insight into key regulators of adaptive fate specification in a progenitor-like landscape. We measured joint gene expression and chromatin accessibility in 23,669 single WT and PRO eNSCs to better understand chromatin changes associated with tumorigenic phenotypes. Although the UMAP reduction of joint gene expression and chromatin accessibility was largely influenced by gene expression differences between WT and PRO eNSCs, there was a clear contribution of mutant-specific accessibility changes in the progenitor-like landscape (Supplementary Fig. 6A). We observed over 13,000 differentially accessible peaks between PRO eNSCs and WT cells with an overall decrease in chromatin accessibility in PRO eNSCs (Supplementary Fig. 6B). When searching these differentially accessible peaks for PRO-specific TF motifs, we found enrichment of early growth and metabolism regulators as well as significant peak changes in neural proliferation and energy maintenance genes, such as *NRG3* (Supplementary Fig. 6C, D). Finally, we leveraged joint multiomic measurements for a combined analysis of TF motif activity using target gene expression (Fig. 3H). We observed significantly elevated activity from several chromatin regulators in PRO eNSCs that specify neural lineages relevant in GBM[28–30]. These included *TCF12*, *ASCL1*, *TCF4*, and *INSM1*. *INSM1* was particularly interesting given the lack of previously described roles in GBM but reported roles in neuronal development and neuroendocrine functions, linking metabolism and neural lineage dynamics[28–30].

## In vivo PRO eNSC tumors recapitulate the cellular heterogeneity in human GBM

We next asked whether the transcriptional dynamics uncovered in vitro were relevant for tumor formation in vivo and how microenvironmental pressures influence cancer cell evolution. We performed scRNA-seq of 23,855 PRO eNSC tumor cells by flow sorting human nuclear antigen+/CD45- cells from xenografted athymic mouse brains at time of median survival (Supplementary Fig. 7A). We first confirmed that PRO tumors demonstrate patterns of human GBM expression using pseudobulk gene set enrichment analysis of TCGA GBM subtypes[31], and observed PRO-dominant developmental signatures in pseudobulk (Fig. 4A) and at the single cell level (Supplementary Fig. 7B). Analysis of single cell state classifications as identified in Neftel et al.[10] showed that the majority of PRO tumor cells were defined as neural progenitor cell- (NPC) or astrocyte cell (AC)-like (Fig. 4B), with a smaller fraction exhibiting Mesenchymal-like signatures, indicating microenvironment-dependent cell state changes during tumor growth. PCA-based UMAP reduction and cell cycle analysis of PRO tumors validated mitotic activity, with all clusters having at least 10–25% actively dividing tumor cells (Supplementary Fig. 7C).

## PRO eNSC tumors transition toward neuronal progenitor phenotypes during evolution

To expand upon our observations of in vitro evolution toward malignancy, we performed trajectory analyses of in vivo PRO eNSC tumors by first using RNA content (CytoTRACE[21]) to order clusters (Fig. 4C, D, and Supplementary Fig. 7D). Inferred tumor dynamics by RNA content began with cluster 1 expression patterns, progressed through the proliferative cluster 9, and ended with the terminal cluster 4 subpopulation. Genes highly correlated with cluster 4 ordering included *INSM1*, *CD24*, *STMN2*, and *SOX4* and were enriched for ontologies related to neuronal development (Fig. 4E, and Supplementary Fig. 7E). When we performed RNA velocity analysis of PRO eNSC tumors, we observed a similar inferred trajectory ending with the terminal cluster 4 subpopulation (Fig. 4F). We performed probabilistic fate mapping in

cluster 4 by modeling transcription initiation, splicing, and degradation over latent time to determine significant cluster-defining gene expression patterns during tumor evolution (Supplementary Fig. 7F). This highlighted dynamic cytoskeleton and microtubule changes before the rapid upregulation of putative lineage regulators *TCF4*, *SOX4*, and *STMN2*.

Having identified putative molecular drivers linking PRO GBM mutations with malignant cell states, we wondered how tumor cell function evolved in vivo in comparison with in vitro serial mutation. We utilized the same ontology-based classification scheme from our in vitro analyses and scored PRO eNSC tumor cells by enrichment of mitochondrial (MTC), glycolytic/ plurimetabolic (GPM), proliferating progenitor (PPR), and neuronal (NEU) functions (Fig. 4G, Supplementary Fig. 7G). Metabolic reprogramming that occurred during in vitro transformation was enriched early in tumor evolution in vivo and then transitioned over time toward an equilibrium between developmental PPR and NEU cell types in more terminal subpopulations, overall suggesting a convergence on the NEU cell state both in vitro and in vivo tumor evolution (Fig. 4G).

## Inhibiting transcriptional drivers of PRO eNSC evolution disrupts neurodevelopmental and oncogenic gene expression and function

We prioritized TFs prominently featured in our evolutionary analyses of PRO eNSCs in vitro and in vivo as putative drivers of metabolic and developmental fate switches during tumorigenesis, including *INSM1*, *TCF4*, and *SOX4*. We first asked how inhibiting these TFs might affect the PRO eNSC gene expression landscape. To address this, we uniquely labeled cells transduced with 2 independent RNA interference (RNAi) hairpin lentiviruses per TF (and control LacZ-targeting RNAi) using individual HTO barcodes for a total of 3 conditions in 3 datasets, one for each gene. Multiplexed sequencing and integrated analyses yielded transcriptomes from 19,098 single cells, and PCA robustly captured the transcriptomic perturbations (Fig. 5A, B, and Supplementary Fig. 8A). Analysis of the PRO RNAi PCA-based UMAP revealed distinct separation of *INSM1* knockdown (INSM1i) cells, a moderate separation of *TCF4* knockdown (TCF4i) cells, and a small fraction of separation between *SOX4* knockdown (SOX4i) and control knockdown PRO eNSCs, which was validated by Spearman similarity correlations between the top variable features at the pseudobulk level (Fig. 5C). In search of commonly deregulated expression patterns between PRO eNSC lineage drivers, we explored the top principal components of the integrated dataset for variation across all genes accounted for by TF expression changes (Supplementary Fig. 8B). While the first principal component captured only the variation in a small, non-specific subpopulation, the second principal component broadly captured the variation across the entire dataset with a robust gradient of change across the horizontal axis between control and TF knockdown conditions. The top genes associated with the second principal component represented a convergent regulatory network between the perturbed TFs and downstream changes related to neuronal development, as represented by *DCX* and *STMN2* in the second principal component.

We next dissected the individual contribution of each TF to gene expression changes in PRO eNSCs. Since *INSM1* RNAi caused the largest perturbation in the PCA-reduced transcription space, we prioritized this chromatin regulator and compared *INSM1* RNAi (*INSM1i*) -disrupted networks with *TCF4* RNAi (*TCF4i*)- and *SOX4* RNAi (*SOX4i*)- dependent variation (Fig. 5D, E, and Supplementary Fig. 9A–C). A comparison of differentially expressed genes within each dataset revealed heterogeneous expression changes related to neuronal progenitor fate commitment in *INSM1i* and *TCF4i* datasets whereas the only significantly altered gene in the *SOX4i* dataset was downregulation of the *SOX4* gene itself (Supplementary Fig. 9D, E). We performed gene ontology analysis of the INSM1-dependent genes to better understand the functional consequences of neuronal lineage-

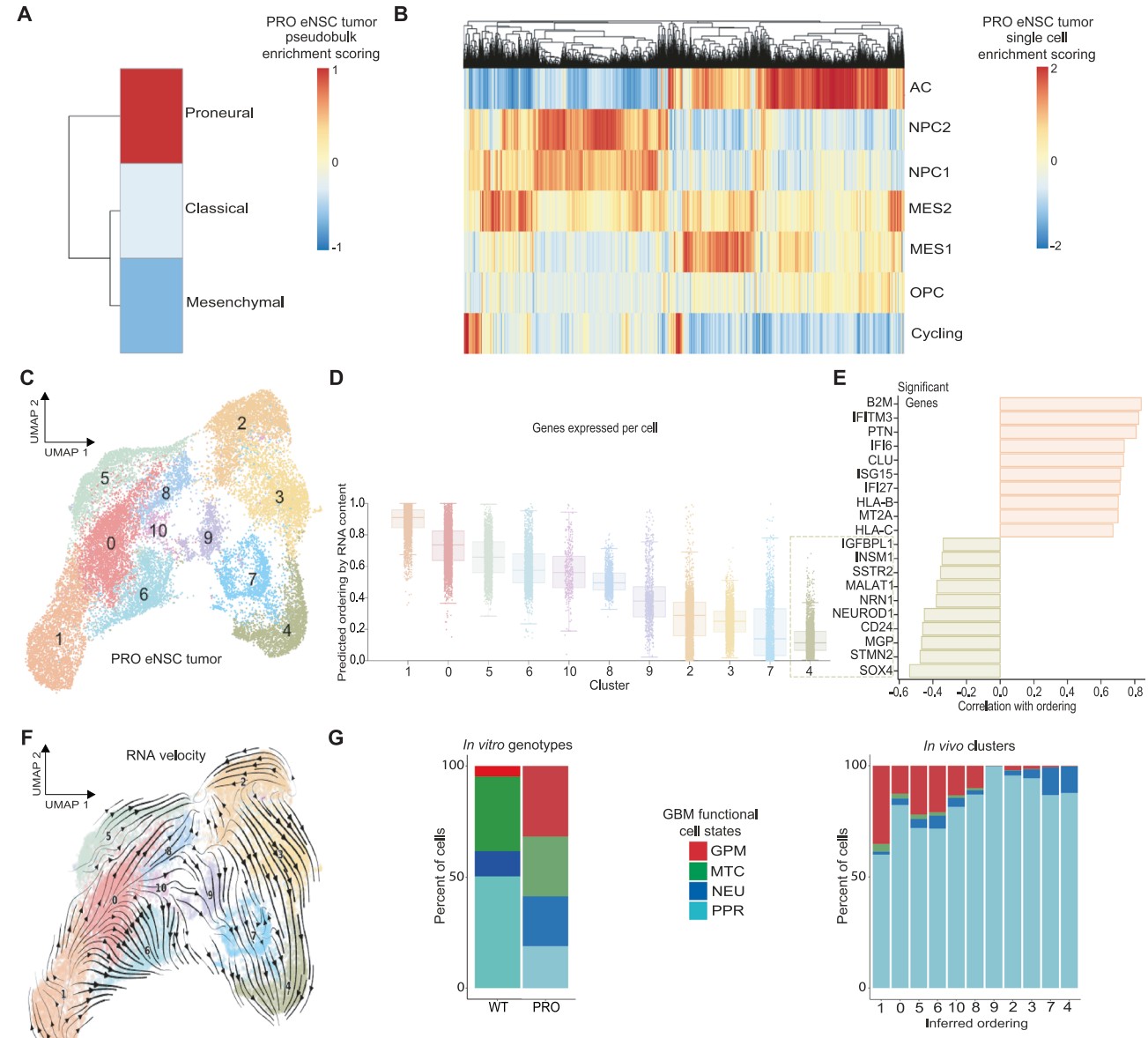

**Fig. 4 | PRO eNSC tumors exhibit metabolic to developmental gene expression changes during evolution. A** Pseudobulk enrichment analysis of PRO eNSC tumors using TCGA GBM bulk subtype gene sets[31]. **B** Single cell enrichment analysis of PRO eNSC tumors by cell state gene sets defined in Neftel et al.[10]. **C** Cluster analysis of PRO eNSC tumor PCA-based UMAP. **D** Inferred ordering of PRO eNSC tumor clusters by RNA content. Box plots indicate the median (center line), interquartile range (hinges), and 1.5x interquartile range (whiskers). **E** Genes most

correlated with the terminal cluster identified by RNA content transcriptional covariance. **F** PRO eNSC UMAP overlaid with RNA velocity vector field. **G** Stacked bar plot displaying proportions of functional GBM subtypes (Garofano et al.[27]) in WT and PRO eNSCs in vitro (left) and in PRO in vivo tumor clusters ordered by inferred trajectory (right), (GPM: glycolytic/plurimetabolic, MTC: mitochondrial, NEU: neuronal, PPR: proliferating progenitor).

related transcriptomic changes and observed significant enrichment of nervous system development, as well as generation of neurons and their signaling pathways (Fig. 5E). This suggests a dependence of PRO eNSCs on INSM1 for epigenetic regulation of neuronal fate commitment in the context of malignant transformation.

To better understand how INSM1 impacts PRO oncogenic gene expression and function in vitro, we first performed a Spearman correlation analysis of WT and PRO mutant eNSCs (combining the triple mutant PRO and RNAi control samples) with the transcriptomic landscape of *INSM1*i PRO eNSCs based on the most highly variable features within the integrated dataset at the pseudobulk level (Fig. 5F). Importantly, INSM1 inhibition in PRO eNSCs caused a reversal of transcriptomic changes towards the non-tumorigenic, WT NSC transcriptional landscape (Supplementary Fig. 9F), suggesting that *INSM1*

RNAi abrogates the malignant phenotype of PRO eNSCs. To validate the functional role of INSM1 in GBM, we first applied the ontology-based GBM classification scheme to *INSM1* RNAi PRO eNSCs in vitro, which demonstrated a substantial decrease in the fraction of glycolytic/plurimetabolic (GPM) cells compared to control PRO cells (Fig. 5G), suggesting that INSM1 also reprograms metabolism in PRO eNSCs. Since PRO eNSCs exhibited enhanced glycolytic activity compared to WT eNSCs (Fig. 3C), we performed Seahorse XF analyses to test if INSM1 is critical for this metabolic phenotype. Indeed, *INSM1* knockdown decreased the glycolytic activity of PRO eNSCs but not that of WT NSCs (Fig. 5H, and Supplementary Fig. 10A).

We then investigated the role of *INSM1* in PRO eNSC malignant transformation by monitoring self-renewal capacity by the extreme limiting dilution assay (Supplementary Fig. 10B-C). Consistent with the

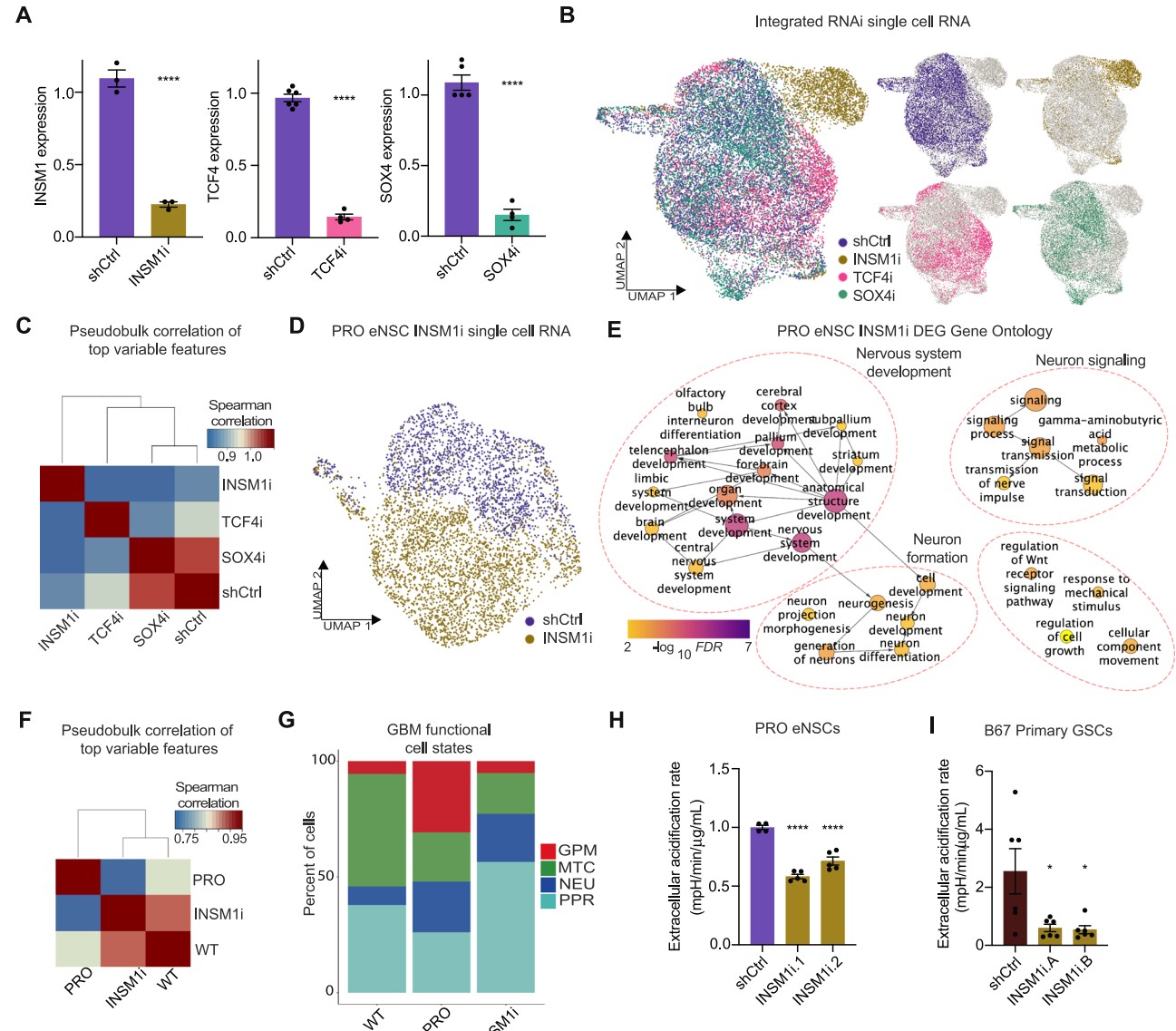

**Fig. 5 | INSM1 inhibition in PRO eNSCs induces metabolic and developmental regression. A** Expression of INSM1, TCF4, or SOX4 mRNA by qRT-PCR normalized to ACTIN and GAPDH expression in PRO eNSCs following inhibition by RNA interference (RNAi). Data represent mean ± SEM in independent replicates (INSM1i and TCF4i: $n = 3$; SOX4i: $n = 4$, Student's t test, two-sided, ****$P < 0.0001$). **B** Single cell transcriptomic profiling of PRO eNSCs following TF inhibition. Shown is the integrated PCA-guided UMAP embedding colored by combined RNAi labels (left) and individual RNAi labels (right). **C** Spearman similarity correlation between shCtrl and TF RNAi in PRO eNSCs by pseudobulk analysis of top variable features. **D** PRO eNSC INSM1i PCA-based UMAP. **E** Gene ontology of PRO eNSC INSM1i differentially expressed genes ($n = 257$). Circle size represents number of genes within each ontology, and color indicates log-transformed FDR.

**F** Spearman similarity correlation between WT, PRO, and INSM1i eNSC by pseudobulk analysis of top variable features. **G** Proportions of GBM functional cell state labeling in WT, PRO, and INSM1i eNSCs (GPM: glycolytic/plurimetabolic, MTC: mitochondrial, NEU: neuronal, PPR: proliferating progenitor). **H** Seahorse XF analysis of glycolysis-dependent extracellular acidification rate in PRO eNSCs following inhibition of INSM1 using two independent RNAi. Data represent mean ± SEM in independent replicates ($n = 4$, ANOVA, ****$P < 0.0001$). **I** Seahorse XF analysis of glycolysis-dependent extracellular acidification rate in B67 primary GSCs following inhibition of INSM1 using two independent RNAi (GIPZ RNAi, Methods). Data represent mean ± SEM in independent replicates ($n = 6$, ANOVA, *$P < 0.05$, **$P < 0.01$). Source data are provided as a Source Data file, including exact statistical analyses and $p$ values.

scRNA-seq analyses, we observed a significant decrease in self-renewal capacity following *INSM1* RNAi in PRO eNSCs but not in WT NSCs, suggesting disruption of the malignant phenotype in PRO eNSCs. In other experiments, we also found that knockdown of *TCF4* and *SOX4* inhibited PRO eNSC self-renewal capacity (Supplementary Fig. 10D), verifying that our integrated analyses of in vitro and in vivo scRNAseq data can successfully identify functional epigenetic drivers of the PRO program. To test the generalizability of INSM1's metabolic and malignant roles in GBM, we used primary glioblastoma stem-like cells (GSCs) as a complementary model system[32,33]. As with the PRO eNSCs, we found that *INSM1* knockdown significantly decreased glycolytic

activity in primary GSCs and significantly decreased self-renewal capacity (Fig. 5I, and Supplementary Fig. 10E, F).

## INSM1 is upregulated in human GBM and associates with the PRO subtype

After confirming that *INSM1* expression regulates neuronal progenitor and metabolic cell states in our PRO eNSC model and primary GBM cells, we wondered whether these changes are relevant in human GBM tumors. To address this, we began by analyzing publicly available TCGA bulk RNA sequencing data from primary GBM patients, as well as available scRNA-seq data from 24,131 cells in 28 human GBM

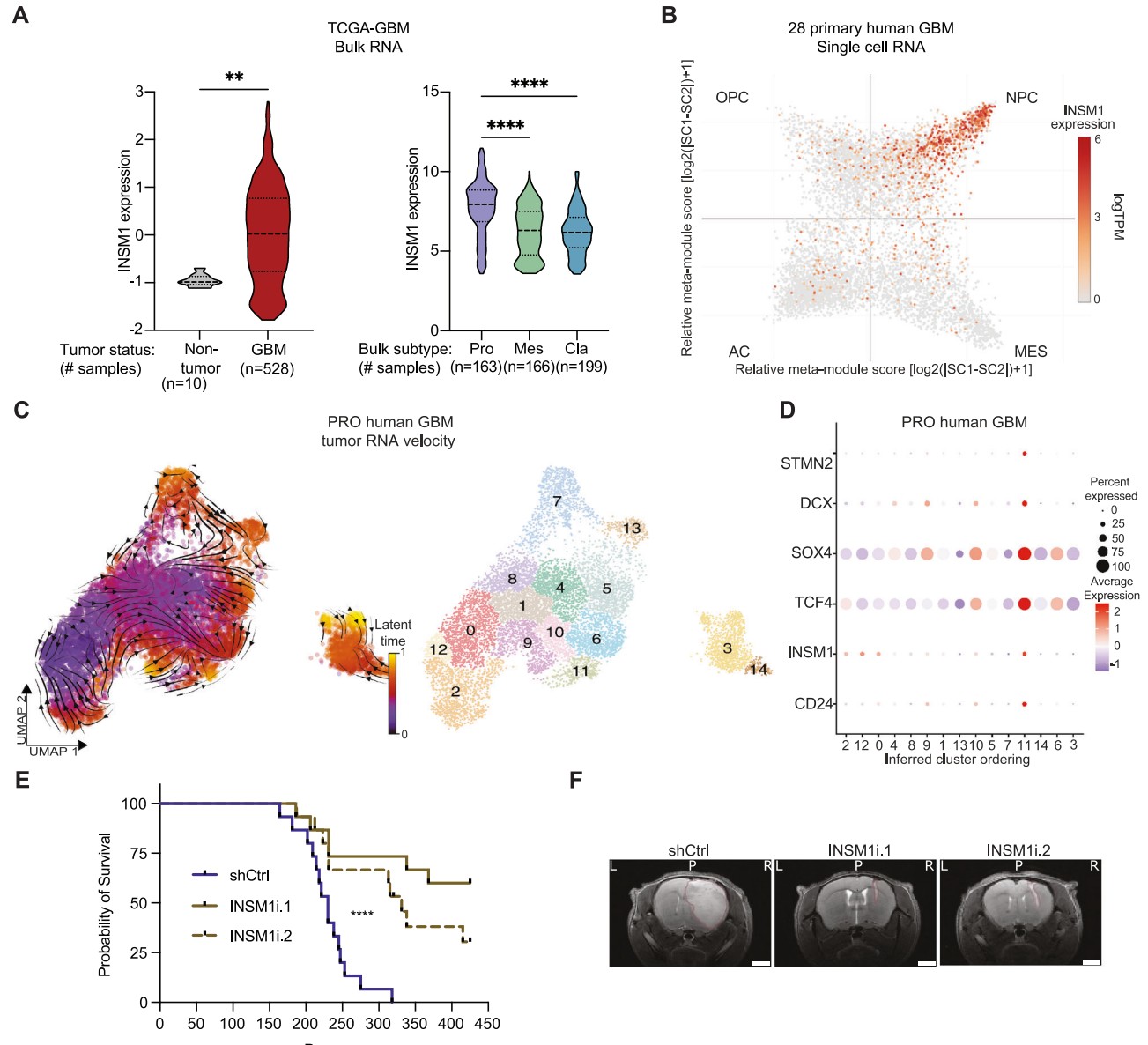

**Fig. 6 | INSM1 is upregulated in human GBM tumors and is critical for in vivo tumorigenicity. A** INSM1 expression in TCGA-GBM bulk RNA-seq data vs. normal brain (left, Student's t test, two-sided, **$P < 0.01$) and GBM bulk classification (right, ANOVA, ****$P < 0.001$). **B** *INSM1* expression in scRNA-seq data from 28 GBM patient tumors[10,31]. Dimension reduction by cell state hierarchy as defined in Neftel et al.[10]. **C** Single cell RNA-seq analysis of a PRO human GBM colored by RNA velocity latent time (left) and clustered subpopulations (right). Shown is the UMAP reduction based on RNA splicing kinetics (Dynamo[22]). **D** Expression of key INSM1-related genes from PRO eNSCs in PRO human GBM clusters ordered by RNA content trajectory inference. **E** Kaplan-Meier survival analysis of mice orthotopically implanted with PRO eNSCs (clones 1 + 2) following lentiviral transduction of two independent INSM1 RNAi (INSM1i) or control RNAi (shCtrl) (clone 1: $n = 10$; clone 2: $n = 5$, log-rank, ****$P < 0.0001$). **F** Representative T2-weighted MRI images of mice from each experimental cohort 200 days post-xenograft with RNAi-transduced PRO eNSCs. Red outlines indicate pathological lesions in the coronal plane of the injection site (scale bar = 1 mm; L = left, P = posterior, R = right). Source data are provided as a Source Data file, including exact statistical analyses and *p* values.

tumors[10,31]. Compared to normal brain, INSM1 is significantly upregulated in GBM and particularly in PRO GBM tumors, compared with Mesenchymal or Classical tumors (Fig. 6A). Cell state analysis of GBM tumor cells, as previously defined[10], revealed a strong association between *INSM1* expression and the neural progenitor cell (NPC)-like state, which is enriched in PRO GBMs (Fig. 6B).

To determine *INSM1* expression dynamics during the evolution of PRO human GBM, we performed scRNA-seq analysis of 11,384 cells from a patient tumor with *PDGFRA* amplification[33]. We first performed gene set enrichment analyses of pseudobulk TCGA signatures and GBM cell state classifications, confirming that our patient tumor displayed PRO and NPC-like expression landscapes (Supplementary

Fig. 11A). To investigate changes in the INSM1 gene expression network during tumor evolution, we first determined the trajectory of tumor subpopulations. Given the heterogeneous nature of human GBM tumors, we began with a high-resolution analysis of splicing dynamics to identify tumor cell subpopulations and cell fate transitions (Fig. 6C). RNA velocity and clustering analyses identified 15 stable subpopulations in our patient sample, of which clusters 3, 9, and 11 were associated with terminal latent time by RNA velocity. Similar to previous analyses, we additionally determined the inferred trajectory of tumor clusters using RNA content-based linear ordering (Supplementary Fig. 11B). This analysis revealed initiation in cluster 2, followed by a similar progression through clusters 9 and 11, and ending with cluster

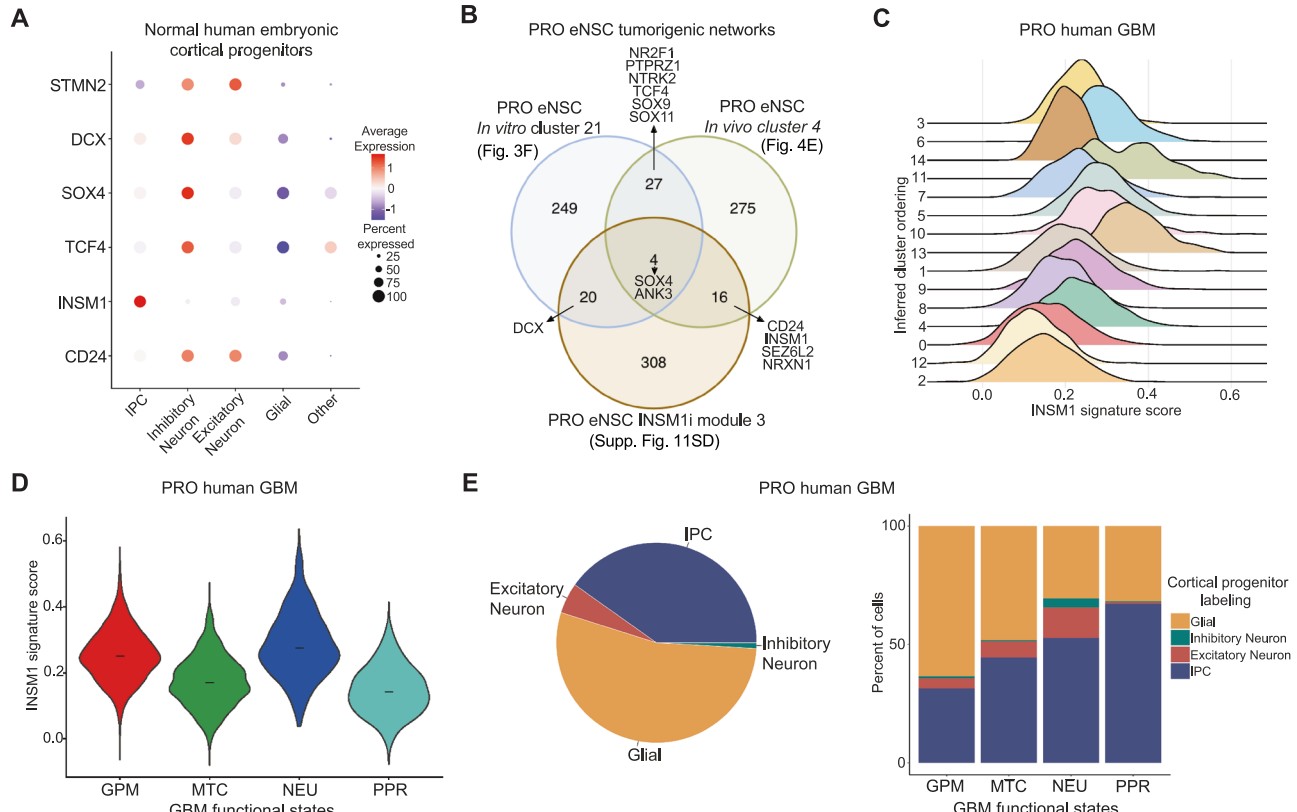

**Fig. 7 | The INSM1 program in PRO human GBM defines a developmental intermediate progenitor cell-like state. A** Expression of key INSM1-related genes from PRO eNSCs in normal human embryonic cortical progenitor (hECP) cell types[34] (IPC: intermediate progenitor cell). **B** Venn diagram comparison of tumorigenic expression networks during PRO eNSC evolution. Circles represent differentially expressed genes from PRO eNSC in vitro cluster 21 (Fig. 3F) and in vivo cluster 4 (Fig. 4G, right), as well as a gene set (module 3) identified by high-dimensional Weighted Gene Co-expression Network Analysis of PRO eNSC INSM1i scRNA-seq data (Supplementary Fig. 11D). 67 shared genes among these analyses

define an oncogenic INSM1 signature (Supplementary Data 1). **C** INSM1 signature scores in PRO human GBM clusters in the scRNA-seq dataset ordered by predicted trajectory. **D** INSM1 signature scores in PRO human GBM tumor cells in the scRNA-seq dataset stratified by classification of GBM functional states. Horizontal lines indicate median scores within each subtype. **E** Pi chart showing proportions of cortical progenitor labeling in the entire PRO human GBM tumor scRNA-seq dataset (left) and proportions of cortical progenitor labeling in PRO human GBM tumor cells stratified by GBM functional states (right).

---

3. Using this predicted trajectory, we analyzed expression of key PRO eNSC INSM1-related genes and found the highest expression of neuronal lineage genes in the more terminal cluster 11 (Fig. 6D).

### Inhibiting INSM1 in PRO eNSCs reduces tumorigenesis in vivo and improves animal survival

Our PRO eNSC GBM model uncovered a neurodevelopmental link between INSM1-dependent malignant transformation and human PRO GBM evolution. We thus directly tested whether INSM1 governs the in vivo tumorigenic potential of PRO eNSCs (Fig. 6E). We orthotopically transplanted *INSM1* knockdown PRO eNSCs using two independent RNAi (vs. control RNAi) into the brains of athymic nude mice. As expected, control PRO eNSCs formed fulminant tumors as seen by MRI (Fig. 6F). Remarkably, *INSM1* RNAi significantly inhibited brain tumor formation by PRO eNSCs (Fig. 6F). Importantly, mice bearing *INSM1* knockdown PRO eNSCs exhibited increased survival compared to mice with control RNAi-infected PRO eNSCs (Fig. 6E). Together, these data indicate that INSM1 is critical for the in vivo tumorigenic capacity of PRO eNSCs, consistent with the in vitro malignant phenotypes observed in our NSC-based GBM model.

### INSM1 regulates the emergence of intermediate progenitor-like and neuronal progenitor-like cells in GBM

We next wanted to understand the relationship between human GBM tumor evolution and normal cortical development with regard to the

INSM1 program. During normal development, INSM1 is specifically expressed in intermediate progenitor cells (IPCs) found in the subventricular zone, which generate inhibitory and excitatory neurons[34,35]. We leveraged a scRNA-seq reference atlas of 121,290 human embryonic cortical progenitor cells (hECPs) from the germinal zone at gestation weeks 15-18, the window of peak neurogenesis[34] (Supplementary Fig. 11C). We analyzed expression of key INSM1-related genes derived from our PRO eNSCs in normal cortical progenitor cells and found that *INSM1* is highly expressed in IPCs, whereas related genes *CD24*, *TCF4*, *SOX4*, *DCX*, and *STMN2* are largely expressed in inhibitory neurons (Fig. 7A, and Supplementary Fig. 11C). We then defined an oncogenic INSM1 signature from the identified mutant-specific transcription networks in our isogenic model (Fig. 7B). First, cluster 21 from our in vitro analyses was selected due to PRO eNSC predominance, as well as TF activity for neuronal specification by joint expression and accessibility analyses (Fig. 3F–H). Second, we chose cluster 4 from the PRO eNSC in vivo analysis (Fig. 4D, E). And third, we performed high-dimensional Weighted Gene Co-expression Network Analysis (WGCNA) of our *INSM1* knockdown PRO eNSC scRNA-seq data (Supplementary Fig. 11D) to identify a relevant INSM1-specific gene set. Altogether, we identified a list of 67 shared genes among these three PRO-specific eNSC networks during evolution to define an oncogenic INSM1 signature (Supplementary Data 1).

When we scored human cortical progenitor cells for expression of our oncogenic INSM1 gene signature, we unexpectedly observed

highest enrichment of the PRO eNSC-derived signature in excitatory neurons and inhibitory neurons in addition to IPCs (Supplementary Fig. 11E). Enrichment analysis of the INSM1 signature in PRO human GBM revealed a gradual increase in enrichment scores along the inferred trajectory with a peak score in cluster 11 (Fig. 7C). The bimodal distribution of scores in cluster 11 suggests further distinction between subpopulations. Clusters 6 and 13 also displayed a relatively high INSM1 score and were found to have hybrid cell functions by gene ontology, which included neuronal functions and metabolic responses to cellular stress (Supplementary Fig. 11F).

We then sought to integrate the oncogenic INSM1 signature, GBM functional cell states, and human cortical progenitor cell types represented in our patient GBM scRNA-seq data. Similar to previous analyses, we stratified our PRO human GBM tumor cells by GBM functional states and observed a transition over predicted cluster ordering from mitochondrial and proliferating progenitor states (MTC and PPR) toward glycolytic/plurimetabolic and neuronal (GPM and NEU) cell states (Supplementary Fig. 11 G). This directly correlated with the evolutionary changes in the oncogenic INSM1 signature, where the NEU tumor cells, followed closely by GPM cells, displayed the highest enrichment of the INSM1 signature, in agreement with our PRO eNSC data (Fig. 7D).

We used reference-based label transfer between the human cortical progenitor cell atlas and our PRO human GBM data to investigate the relationship between normal developmental cell types and INSM1-dependent GBM tumor dynamics. Overall, the human GBM cells largely consisted of glial- and IPC-like cells with a small proportion of both excitatory neuron- and inhibitory neuron-like cells (Fig. 7E). Cluster 11, which displayed the highest INSM1 signature enrichment, was comprised of almost 50% IPC-like cells and interestingly had the largest proportion of inhibitory neuron-like cells, although this represented a small fraction of the total cluster (Supplementary Fig. 11G). By correlating INSM1 signature changes and cortical progenitor cell type labels between GBM functional states, we observed that the increase in INSM1 signature score from GPM to NEU cell states was accompanied by the expansion of IPC-, excitatory neuron- and inhibitory neuron-like cells (Fig. 7E). This highlights the importance of increased INSM1 expression in late PRO human GBM tumor cell evolution and suggests the emergence and hijacking of an IPC-like cell state.

## Discussion

By leveraging targeted CRISPR/Cas9 editing in human ESCs, we developed a NSC model of PRO GBM to understand epigenetic evolution during tumor formation. Orthotopic injection of PRO eNSCs led to tumor formation in mice with histopathology characteristic of GBM. Clonal differences between our PRO genotypes may be due to differences in *TP53* loss of heterozygosity (LOH). Typically, wildtype TP53 regulates its own protein expression by upregulating transcription of the ubiquitin-ligase MDM2, which targets TP53 for degradation[36,37]. Thus, the timing of *TP53* LOH in a given clone may potentially explain why we see malignant growth in a fraction of mice injected with double mutant eNSCs. In addition, *TP53* LOH likely increases genome instability and accelerates selection of oncogenic drivers that promote malignant growth. Thus, we have opportunities to study various contributions of genetic and epigenetic evolution in our eNSC model. Interestingly, *TP53* LOF is responsible for increased *TERT* expression in the context of the TERT*p* mutation in our eNSCs, and *TP53* mutation significantly correlated with increased *TERT* expression in GBM. Although there is a TP53 binding site within 1 kb upstream of the *TERT* promoter, TP53 is also known to antagonize the effects of SP1, a potent activator of the mutant *TERT* promoter[38]. In preliminary experiments (Supplementary Fig. 3H), we have found increased GABPβ1 mRNA expression in eNSCs with both TERT*p* and TP53 mutation compared to single *TERT*p mutant, suggesting a possible increase in GABP-mediated *TERT* transactivation, but this remains to be further studied[16,17]. The mechanism by which oncogenic activation of the mutant *TERT*

promoter, both with and without supporting mutations, affects telomere maintenance in our PRO model is an important open question.

Leveraging joint multiomic profiling of our eNSCs in vitro enabled high-resolution network mapping of TFs controlling fate specification in mutant eNSCs, which we validated through orthogonal splicing-based lineage trajectory analyses of PRO eNSC tumors that formed in vivo. We identified chromatin regulators that consistently appeared as significant expression changes during transcriptome evolution (Supplementary Fig. 6E–G), including *SOX4* and *TCF4*, which have been implicated in invasive signatures of GBM[12]. Of particular interest, INSM1 has been demonstrated to play a defining role in the identity of IPCs during development and their delamination from the subventricular zone to generate outer cortical neurons[38,39]. While mounting evidence suggests that normal neurodevelopment mechanisms are hijacked by GBM cells[12,40,41], it is important to note that trajectory analysis based on scRNA-seq data (*e.g.*, CytoTRACE and Dynamo) relies on the assumption that cancer progression obeys principles of physiologic RNA dynamics. Alternative methods (*e.g.*, Monocle3[24] or Slingshot[42]) infer trajectory from models constructed following dimensional reduction, as opposed to biological metrics, such as number of genes expressed or RNA splicing, utilized by CytoTRACE and Dynamo, respectively. Though preliminary analyses with these methods validate the biological significance of modeling RNA dynamics in our data (Supplementary Fig. 5A), future studies analyzing these datasets using alternative analyses could provide new insights and reduce certain technical or biological variations. For example, downsampling in scRNA-seq can be useful for standardization but inherently discards data, which can reduce sensitivity for detecting rare populations, obscure subtle biological differences, and introduce stochastic variation due to the random removal of reads or cells. We chose not to downsample in this study to avoid these issues. Nevertheless, we leveraged our transcriptomic analyses in eNSCs to define an oncogenic INSM1 signature that exhibited upregulation in later cycling subpopulations during evolution of a human PRO GBM tumor.

To complement the developmental framework of our isogenic stem cell model, we investigated the association between oncogenic INSM1-related genes, GBM functional states, and developmental human cortical progenitor cell types. During human cortical development, peak neurogenesis occurs between gestation weeks 15–18 when radial glia in the subventricular zone undergo asymmetric division to produce one radial glia and an intermediate progenitor cell[34]. *Insm1* knockout in mouse cortical progenitors disrupts normal brain development by disrupting the balance between proliferating progenitors and terminally differentiated neurons[30]. In mice, the lineage commitment toward IPCs is regulated by a transition in TF-mediated chromatin states from Pax6- to Eomes-dependent networks[39]. The balance between self-renewing progenitors and commitment toward excitatory and inhibitory neuronal progenitors is regulated by feedback networks between Pax6, Eomes, Insm1, and Dlx2. While inhibitory interneurons are largely formed and migrate from the ganglionic eminence during cortical development, it was recently found that IPCs in the subventricular zone, where *Insm1* is typically expressed, are capable of generating inhibitory neurons[35]. While gene set enrichment analysis of the oncogenic INSM1 signature derived from PRO eNSC transcriptome evolution displayed enrichment in normal IPCs, excitatory and inhibitory neurons unexpectedly displayed the highest enrichment scores for the oncogenic INSM1 signature. This highlights the deregulation of normal developmental processes for tumor-specific evolution and plasticity.

Importantly, we found *INSM1* mRNA is highly expressed in GBM from The Cancer Genome Atlas (TCGA). *INSM1* expression is significantly associated with bulk and single cell mRNA PRO GBM signatures but is also elevated in Classical and Mesenchymal tumors. In other reports, INSM1 was detected in 80-88% of GBM tumors by immunohistochemistry, suggesting that this TF may be more generally

relevant for GBM biology[29,43]. INSM1 is a particularly interesting candidate for therapeutic targeting in GBM given its small window of expression during development and low or absent expression in many tissues[29]. Therefore, INSM1 may represent a promising GBM-specific target that inhibits crucial cell fate transitions during tumor evolution and growth.

Thus, we have demonstrated that our human ESC-derived model of PRO GBM accurately recapitulates molecular and cellular phenotypes of patient tumors and serves as a robust platform for understanding genetic and epigenetic evolution of isogenic GBM-like cells. How this model responds to and evolves with therapeutic stress, such as chemotherapy or radiation therapy, remains an important future question. In addition, identifying the direct transcriptional targets of INSM1 and its regulatory mechanisms in GBM may provide therapeutic opportunities for suppressing tumor growth, invasion, and recurrence.

## Methods

### Institutional approvals
All research in this study complies with the relevant Washington University School of Medicine ethical regulatory committees: Institutional Review Board (IRB#201211019, IRB#201111001 #201709124), Embryonic Stem Cell Research Oversight Committee (ESCRO# 17-005), and Institutional Animal Care and Use Committee (IACUC) (#24-0047). All participants donating tissue signed informed consent prior to tissue banking (IRB#201111001). Human embryonic stem cell line H1 (NIH registry #: 0043) was purchased from WiCell. Our animal protocol adheres to NIH and American Association for Laboratory Animal Science (AALAS) guidelines.

### Generation of genetically engineered hESC clones
The wildtype (2E1, 2F9), single mutant *TERT*p (2A7, C7), and double mutant *TERT*p + *TP53* (7F8, 6C4) Proneural cell lines were created by the Genome Engineering & Stem Cell Center (GESC@MGI) at Washington University in St. Louis. Briefly, synthetic gRNA targeting the sequence and donor ssODNs for knock-in were purchased from IDT, complexed with Cas9 recombinant protein and the ribonucleoprotein was co-transfected into the H1 hESCs[44,45]. The transfected cells were single cell sorted into 96-well plates, and clones from single cells were identified using next generation sequencing to analyze the target site region as those harboring knock-in mutation. Mutations were finally validated by Next Generation and Sanger sequencing.

TERT promoter −124C > T gRNA: 5′-gggctgggagggcccggaggngg

ssODN:

5′-cgcctcctccgcgcggaccccgccccgtcccgacccctcccgggtccccggcc-
cagccccttccgggccctcc-
cagcccctcccttcctttccgcggccccgccctctcctcgcggcgcga

TP53 exon 6 743 G > A gRNA: 5′-gcatgggcggcatgaaccggngg

ssODN:

5′-gactgtaccaccatccactacaactacatgtgtaa-
cagttcctgcatgggcggcatgaaccagagacccatcctcaccatcatcacactggaa-
gactccaggtcaggagccacttgccaccctgca

### Cell culture
Proneural mutant cell lines (2E1, 2F9, 2A7, C7, 7F8, 6C4) derived from H1 hESCs (male) were cultured on plates coated with Matrigel hESC-Qualified Matrix (Corning #8774552) in mTeSR1 media (Stemcell Technologies #85850) supplemented with ROCKi (ThermoFisher #A2644501) during harvesting, passaging, and plating of experimental cultures. NSCs were cultured on poly-L-ornithine (Sigma-Aldrich #P2533) and laminin-coated (Sigma-Aldrich #L2020) plates in NSC maintenance media consisting of DMEM/F12 with GlutaMAX (Gibco #11330-032), D-glucose (Sigma-Aldrich #G8270), Insulin (Sigma-Aldrich #SLBZ8936), 1x N-2 supplement (Gibco #17502-048), 1x B-27 supplement (Gibco #12587-010), and 20 ng/mL hEGF (Peprotech #AF-100-15) and mFGF (Peprotech #100-18B). Growth factors (EGF/FGF-2)

were replenished every 2-3 days. Cells were routinely used between passages 5 and 20. Informed consent was obtained from patients for use of human tissue and cells, and all human tissue-related protocols used in this study were approved by the Institutional Review Board (Washington University). Human embryonic kidney 293 T (HEK293T) cells were cultured in DMEM with 10% fetal bovine serum (Fisher #10082-147) and penicillin/streptomycin (Life Technologies #15140122). All cell lines were incubated at 37 °C with 5% $CO_2$. Lentiviral transduction was performed by adding virus with 5 μg/mL of polybrene for 6 h to cells. Cells were selected for 5-7 days in 5 μg/mL of blasticidin (Sigma-Aldrich #15205) for expression of mutant PDGFRA and in 1 μg/mL of puromycin (Sigma-Aldrich #P8833) for 1-2 days after infection of RNAi constructs. For self-renewal, clonogenic assays, and in vivo tumorigenicity experiments, NSCs were utilized 6 days following indicated RNAi lentiviral infections.

### Neural stem cell differentiation
Generation of neural stem cells was adapted from a previous study to an adherent, monolayer format of neuroectodermal differentiation[14]. Proneural mutant ESCs at 95–100% confluency were dissociated using TrypLE (Gibco #12604-013) and seeded on Matrigel-coated plates in mTeSR1 media using $5 - 6 \times 106$ cells per line. Following 24 h (Day 1), media was replaced with KSR media containing DMEM/F12 with L-glutamine (Gibco #11330-032), Knockout Serum Replacement (Gibco #10828028), non-essential amino acids (NEAA #Corning, 25-025), and β-mercaptoethanol (Gibco #2020-11-30), supplemented with small molecules 125 ng/mL Noggin (R&D Systems #6057-NG), 10 mM SB431542 (StemCell Technologies #72234), and 200 nM LDN193189 (Reprocell #04-0074). KSR media was replaced with fresh KSR media on days 2 and 3. On day 5, KSR media was replaced with KSR and N2 media (DMEM/F12 with Glutamax, N2 supplement, and NEAA) at a 3:1 ratio plus small molecules Noggin, SB431542, and LDN193189. On days 7 and 9, media was replaced with KSR + N2 media at 1:1 and 1:3 ratios, respectively, supplemented with the same small molecules. On day 11, media was replaced with NSC maintenance media supplemented with small molecules and the following day, cells were harvested (TrypLE for 30 min. at 37 °C) and split into two cultures on Matrigel-coated plates in NSC maintenance media and ROCKi – these were considered P0 NSCs. During passages 1-5, newly differentiated NSCs were expanded, harvested, and plated for validation of stem cell markers and banking of frozen stock cultures. Validated NSCs were transduced with lentiviral PDGFRA constructs between passages 5 and 7.

Stem cell marker validation qRT-PCR primers
Gapdh-f: 5′-AATTTGGCTACAGCAACAGGGTGG
Gapdh-r: 5′-TTGATGGTACATGACAAGGTGCGG
β-actin-f: 5′-ATGATATCGCCGCGCTCGTCGTC
β-actin-r: 5′-TGACCCATGCCCACCATCACG
Oct4-f: 5′-AGAACATGTGTAAGCTGCGG
Oct4-r: 5′-GTTGCCTCTCACTCGGTTC
Pax6-f: 5′-GCCCTCACAAACACCTACAG
Pax6-r: 5′-TCATAACTCCGCCCATTCAC

### Immunocytochemistry
Cells were plated at 30 K cells/well (NSCs) or 60 K cells/well (ESCs) in a Matrigel-coated black 96-well plate and fixed with 4% PFA (Electron Microscopy Science #15710-S) for 30 min at room temp. Cells were blocked and permeabilized with ICC solution consisting of 5% normal donkey serum (Sigma-Aldrich #39663) and 0.1% Triton-X (Sigma-Aldrich #X100) for 45 min. at room temp. and incubated with Oct-4 (Cell Signaling Technology #2750S, 1:400) or Pax-6 antibodies (Cell Signaling Technology #60433S, 1:200) overnight at 4 °C. Cells were washed with ICC solution and incubated with Anti-rabbit antibody conjugated with a 488 nm fluorophore (Fisher Scientific #A11034) for 1 h at room temp. Cells were washed with ICC solution and incubated with Hoechst 33258 (Fisher Scientific #H3570) prior to imaging.

### Transient transfection

Polyethyleneimine (PEI, Polysciences #24765-2) was dissolved in ddH2O to a concentration of 1 mg/mL. The solution was the adjusted pH 7.0 and filter sterilized. A mixture of plasmid DNA and PEI solution was made in OPTI-MEM (ThermoFisher #31985062) and incubated at room temperature for 20 min. DNA/PEI complexes were applied to cells, and media was changed after 12-16 h.

### Lentiviral Production

HEK293T cells were plated with a goal density of 70-80% after 1 day. The next day, transfection was performed using the PEI transfection method to introduce the plasmid of interest along with packaging plasmid psPAX2 and envelope plasmid pCMV-VSVG to OPTI-MEM. Media was replaced the following day and on day 6, medium from the plates was collected and spun down at 1200x$g$ for 5 min. at 4 °C. The supernatant was filtered through 0.45-micron filters. Lenti-X Concentrator (Clontech #631231) was added to the filtrate and mixed, and the tubes were incubated at 4 °C for 16 h. Lentiviruses were then centrifuged at 1500x$g$ for 45 min. at 4 °C. The supernatant was aspirated; pellets were resuspended in one-tenth of the original medium volume of cold PBS and stored at −80 °C in aliquots. Viral copy number was adjusted for transduction of NSCs on the basis of titer measured using the Lenti-X qRT-PCR titration kit (Clontech #631235).

### Immunoblotting

Cells were lysed in 1% NP-40 lysis buffer containing 20 mM Tris [pH 8], 200 mM NaCl, 10% glycerol, 1 mM EDTA, 10 mM NaF, 1 mM sodium orthovanadate, and a protease inhibitor cocktail (Sigma-Aldrich #539131). Samples were separated by SDS-PAGE and transferred to 0.45 µM Immobilon-P PVDF membrane (EMD Millipore #IPVH00010). Membranes were blocked in 5% Milk in Tris-buffered saline with Tween-20 (TBST) at room temperature and incubated with primary antibodies at 4°C overnight. Membranes were washed with TBST and incubated with appropriate horseradish peroxidase-conjugated secondary antibodies for 1 h at room temperature. Membranes were then washed with TBST and developed using Pierce ECL western blotting substrate (Thermo Scientific #32106). Antibodies used include PDGFRA (CST, #3174S, 1:1000), phospho-PDGFRA (Y754, Abcam #ab5460, 1:750), phospho-ERK (T202/4, CST #9101S, 1:1000), phospho-AKT (S473, CST #4060S, 1:1000), TP53 (CST #9282, 1:500), and β-ACTIN (Santa Cruz #sc-47778, 1:2000).

### Extreme limiting dilution analysis

Cells were plated at five-fold dilutions (3000, 600, 120, 24, 5, or 1 cell/well) in Corning ultralow attachment 96-well plates. Fourteen days later, the number of wells containing spheres were counted and used to calculate the frequency of self-renewing NSCs by online software (http://bioinf.wehi.edu.au/software/elda/).

### Soft agar colony formation assay

Cells were plated in 0.3% agarose (Lonza #50101) in NSC maintenance media on top of 0.6% agarose/NSC media-coated plates at a density of 7,500 cells/well. Every 3 days, 300 µL fresh media was added to each well, and after 21 days, media was carefully aspirated and each well was stained with 500 µL of 0.1% crystal violet solution (Sigma-Aldrich #C0775) for 30 min at room temp. Following 5x ddH2O washes, plates were imaged and colonies were counted using ImageJ software following 8-bit conversion and threshold adjustment.

### Xenotransplantation

6-8 week old female athymic nude mice ages (CrTac:NCr-Foxn1nu, Taconic Biosciences) were used for this study. Mice were housed in static microisolator caging on 1/8 inch corncob bedding with ad libitum access to Lab Diet 5053 chow and autoclaved water. Temperature was maintained between 70° +/− 2° F, humidity within 30–70%, and 12:12 h dark to light cycle. All mice were free of Pneumonia Virus of Mice (PVM), Reovirus 3 (REO3), Sendai virus, Mycoplasma pulmonis, Minute Virus of Mice (MVM), Theiler's Murine Encephalomyelitis Virus (GDVII), Lymphocytic Choriomeningitis Virus (LCMV), Polyoma Virus, Mouse Rotavirus (EDIM), Ectromelia Virus (Mousepox), Mouse Adenovirus, K Virus, Mouse Parvovirus (MPV), Cytomegalovirus, Mouse Hepatitis Virus (MHV), Clostridium piliforme, Streptococcus pneumonia, Bordetella bronchiseptica, Streptobacillus moniliformis, Corynebacterium kutcheri, Salmonella spp. Citrobacter rodentium, murine pinworms and fur mites. Animals were used in accordance with a protocol (#21-0083) approved by the Animals Studies Committee of the Washington University School of Medicine per the recommendations of the Guide for the Care and Use of Laboratory Animals (NIH). ~50,000 cells per animal were injected stereotactically into the right putamen of ~6-week-old female athymic nude mice. The coordinates used were: 1 mm rostral to bregma, 2 mm lateral, and 2.5 mm deep. Mice were euthanized if they were unable to feed, unable to walk, developed a seizure disorder, or if body weight loss exceeded 20% of original body weight or 20% less than age and sex-matched normal controls. Additionally, if the tumor itself ulcerated through the skull or if there was evidence of obvious local infections, then animals were euthanized. All unforeseen health concerns were discussed with veterinary staff with regard to the need for euthanasia.

### Small animal MRI

MRI experiments were performed on a Bruker BioSpec 9.4 T MRI (Bruker, Billerica, MA) using a 86 mm ID volume transmitter coil and a 4-channel mouse brain CryoProbe array receiver coil. Animals were anesthetized with 1–1.5% isoflurane (and body temperature was maintained using a circulating warm water pad). Fat-suppressed T2-weighted RARE images were acquired of the brain using the following parameters: TR = 2500 s, TE = 33 s, matrix size = 256 × 256, spatial resolution = 700 µm, 9 slices at 5 mm thickness, rare factor = 9, 1 average, acquisition time = 1 min. Images were analyzed and tumor volumes extracted using the semi-automatic segmentation analysis software ClinicalVolumes (ClinicalVolumes, London, UK).

### Histology

Mice injected with Proneural eNSCs were perfused, brains harvested, fixed with 4% PFA, cryopreserved with 30% sucrose, flash-frozen, and sectioned at 10 µm thickness along the coronal plane using the tissue near the needle tract for cell injection based on stereotaxic coordinates. Brain sections were washed with PBS, blocked with BSA and goat serum solution, and membrane permeabilized with 0.3% Triton-X in blocking buffer. Sections were then incubated overnight at 4 °C with primary antibody against human nuclear antigen (Novus Biologicals #NBP2-34342, 1:500) and 2 h at room temp. with fluorophore-conjugated anti-mouse secondary antibody (ThermoFisher #A-21202). Sections were counterstained with Hoechst 33258 (Sigma #94403) 5 min at room temp. and mounted with Fluoromount-G (Southern Biotech #0100-01).

### TERT expression qRT-PCR primers

TERT expression was measured using the TaqMan detection reagent (ThermoFisher #4440040) and the following primers:
 ACTIN (ThermoFisher #Hs99999903_m1)
 GAPDH (ThermoFisher #Hs99999905_m1)
 TERT (ThermoFisher #Hs00972650_m1)

### Telomere repeat amplification protocol

Protein was extracted from frozen cell pellets using NP-40 buffer (25 mM Hepes, 150 mM KCl, 1.5 mM MgCl2, 0.5% IGEPAL CA-630, and 10% glycerol, pH 7.5, in RNase-free water) supplemented with Pierce Protease inhibitor (ThermoFisher #A32955), PhosSTOP (Roche #04906837001), and RNase OUT (Invitrogen #P/N-51535) to a final

concentration of 40 U/ml in the NP-40 protein extraction buffer. Samples were incubated on ice for 15 min and centrifuged at 12,000*xg* for 10 min at 4 °C, and the supernatant was transferred to a clean tube placed on ice. Protein quantification was performed using the Pierce BCA Protein Assay Kit (ThermoFisher #23225) according to the manufacturer's instruction. Telomere extension reactions were performed using 2.0, 0.5, and 0.125 µg of protein extract per sample, and the resulting products were amplified by PCR following a two-step TRAP protocol from the manufacturer (TRAPeze Telomerase Detection Kit, Millipore #S7700). PCR products were resolved on a 9% polyacrylamide 37.5:1 gel at 250 V for 3 h. Gel was dried on a gel dryer (Bio-Rad #583) at 80 °C for 30 min, exposed for 15 min to a Kodak BioMax MR film (Carestream #8701302) at −80 °C, and developed.

## CUT&RUN

A magnetic bead-based protocol was used[46]. Briefly, 5 × 105 cells were harvested by centrifugation (600*xg*, 3 min in a swinging bucket rotor) and washed in ice cold phosphate-buffered saline (PBS). Cells were bound to concanavalin A magnetic beads (Bang Laboratories #BP531) and permeabilized with digitonin (Sigma-Aldrich #300410). Cells were briefly washed in 1.5 ml Buffer 1 (20 mM HEPES pH 7.5; 150 mM NaCl; 2 mM EDTA; 0.5 mM Spermidine; 0.1% BSA) and then washed in 1.5 ml Buffer 2 (20 mM HEPES pH 7.5; 150 mM NaCl; 0.5 mM Spermidine; 0.1% BSA). Cells were resuspended in 500 µl Buffer 2 and 3 µl antibody was added and incubated at 4 °C for 16 h. Cells were washed 3 x in 1 ml Buffer 2 to remove unbound antibody. Cells were resuspended in 300 µl Buffer 2 and 5 µl protein A-conjugated micrococcal nuclease (pA-MN) was added and incubated at 4 °C for 4 h. Nuclei were washed 3 x in 0.5 ml Buffer 2 to remove unbound pA-MN. Tubes were placed in a metal block in ice-water and quickly mixed with 100 mM $CaCl_2$ to a final concentration of 2 mM. The reaction was quenched by the addition of EDTA and EGTA to a final concentration of 10 mM and 20 mM, respectively. Cleaved fragments were liberated into the supernatant by mixing at 500 rpm and 37 °C for 10 min. DNA fragments were extracted from the supernatant using QIAquick Spin columns (QIAGEN) and used for the qRT-PCR. Antibodies used include anti-trimethyl-histone H3 Lys4 (Millipore #05-745 R), anti-histone H3 monomethyl K4 (Abcam #ab8895), and anti-trimethyl-histone H3 Lys9 (Millipore #07-442).

## Chromatin immunoprecipitation

Ten million cells were harvested, cross-linked with 1% formaldehyde for 10 min, and quenched with 0.125 M glycine for 5 min. Cells were lysed in 1 mL RIPA buffer (10 mM Tris-HCl, 1 mM EDTA, 0.1% sodium deoxycholate, 0.1% SDS, 1% Triton X-100, pH 8.0), and rotated for 15 min at 4 °C. Cell lysates were centrifuged at 2,300*xg* for 5 min at 4 °C to isolate the nuclei. Nuclei were suspended in 500 µL of 0.5% SDS lysis buffer (0.5% SDS, 10 mM EDTA, 50 mM Tris-HCl, pH 8.0) and subjected for sonication to shear chromatin fragments to an average size between 200 bp and 500 bp on the Covaris E220 focused-ultrasonicator (peak power = 140, cycles/burst = 200, duty factor= 20, duration = 300 s). Fragmented chromatin was centrifuged at 16,100*xg* for 10 min at 4 °C, 450 µL of supernatant was transferred to a new Eppendorf tube, and NaCl was added to a final concentration of 300 mM. We preincubated the ChIP-grade antibodies with 10 µL of MyOne Streptavidin T1 Dynabeads (ThermoFisher #65601) by rotation for 30 min at 20-25 °C and then isolated antibody-bound beads by magnet before adding sonicated lysates. Supernatant was then incubated and rotated with antibody-bound magnetic beads at 4 °C overnight. After overnight incubation, Dynabeads were washed twice with 1 mL of 2% SDS, twice with 1 mL of RIPA buffer with 0.5 M NaCl, twice with 1 mL of LiCl buffer (250 mM LiCl, 0.5% NP-40, 0.5% sodium deoxycholate, 1 mM EDTA and 10 mM Tris-HCl, pH 8.0), and twice with 1 mL of TE buffer (10 mM Tris-HCl, 1 mM EDTA, pH 8.0). The chromatin was eluted in SDS elution buffer (1% SDS, 10 mM EDTA, 50 mM Tris-HCl, pH 8.0) followed by reverse cross-linking at 65 °C overnight. The ChIP DNA was treated with

RNase A (5 µg/ml) and protease K (0.2 mg/ml) at 37 °C for 30 min, and purified using QIAquick Spin columns (QIAGEN). ChIP DNA was analyzed by qRT-PCR. Antibodies used include GABPα (Santa Cruz #sc-28312) and GABPβ1 (Proteintech #12597-1-AP).

## TP53 RNAi and qPCR primers

Inhibition of TP53 was performed using a previously published and validated RNAi targeting the following sequence[19,20]:

 5′- GACTCCAGTGGTAATCTACT

 TP53 expression was measured using the following primers:

 TP53-f: 5′-CAGCACATGACGGAGGTTGT

 TP53-r: 5′-TCATCCAAATACTCCACACGC

## Analysis of TP53 mutants and TERT expression in TCGA

TCGA-GBM data pre-analyzed for telomere maintenance-related genetic variants, TERT expression, and telomere length was downloaded[47]. Samples were filtered to include only the GBM disease type, exclude samples where TERT expression was 0 or not detected, include samples with TP53 mutations annotated as binarized wildtype (0) or mutant (1), and exclude samples with ATRX, TERC, or DAXX mutations or samples with TERT fusions.

## Single cell preparation and multiplexed sequencing

Single-cell RNA sequencing was performed as reported[33]. Cells were single cell dispersed in TrypLE and stained with hashtags according to the manufacturer protocol. Each of the 4 Proneural genotypes were labeled separately and pooled for submission to Washington University School of Medicine Genome Technology Access Center for library preparation and sequencing using the Chromium Single Cell 3′ Library and Gel Bead Kit v3 (10X Genomics). Cells were sequenced on the NovaSeq 6000 (Illumina) at 2x150bp. For the feature library, we used custom 8 bp dual same TotalSeqA-HTO indexes (TruSeqIDTdual-D807, CAGATCAT-CAGATCAT) and IDT HTO cDNA PCR additives.

## Multiplexed sequencing analysis workflow

Alignment and transcript quantification were performed using the Cell Ranger pipeline (10X Genomics, default settings, version 3.0.1). Using the Seurat package v4 in R, samples were demultiplexed and cells containing fewer than 1,000 expressed genes, more than 6,000 genes, or containing more than 10% mitochondrial transcripts were removed. Genes that were expressed in fewer than three cells were also removed, resulting in 11,404 cells from 4 samples with a mean of 10,517 reads/cell over an average of 3,177 genes (WT: 3,390 cells, TERTp: 2,598 cells, TERTp+TP53: 2492 cells, PRO: 2924 cells). For each cell, the expression of each gene was normalized to the sequencing depth of the cell, scaled to a constant depth (1 × 106), and log-transformed. RNA content analyses were performed using the CytoTRACE library in R using the normalized gene expression matrix and sample labels. CytoTRACE determines which genes' expression are most correlated with the gene counts signature and identifies association with initial populations when scores > 0 while scores <0 associate with terminal populations. Using genes with a correlation score < −0.05 we analyzed the resultant 405 genes for pathway ontologies using the BiNGO app v3.0.5 in Cytoscape visualization platform v3.9.1.

 RNA velocity analyses were performed using velocyto v0.17.16 scVelo v0.2.5, and Dynamo v1.1.0 in Python v3. Spliced and unspliced transcript counts were determined using the velocyto command line interface and the output loom file was further processed using scVelo for dynamical modeling, Monocle3 v3.21 and CellRank v1.5.1 for pseudotime analysis and single-cell fate mapping, and Dynamo for an analytical model of differential geometry.

## Mitochondrial respiration studies

eNSCs were seeded at 6 × 104 cells/well on Seahorse cell culture plates (Agilent, 101085-004) overnight. Media was replaced with DMEM

(Agilent, 103575-100) supplemented with 2.5 mM glutamine, 17.5 mM glucose and 1 mM sodium pyruvate and the plate was placed in a non-$CO_2$ incubator at 37 °C for 1 h. The cell culture plate was placed in a Seahorse XFe96 Analyzer. Sequential injections of 3 μM oligomycin, 0.25 μM carbonyl cyande-4-(trifluoromethoxy) phenylhydrazone (FCCP), and 1 μM rotenone and 2 μM antimycin A were placed in the analyzer injection ports. All compounds were from a Seahorse XF Cell Mito Stress Test Kit (Agilent, 103015-100). Four ECAR and OCR measurements were recorded for baseline and following each injection. Protein quantification was performed using the Pierce BCA Protein Assay Kit (ThermoFisher #23225) according to the manufacturer's instructions for normalization of each well's measurements.

### Single nuclei sample preparation and sequencing

Cells were harvested for delivery to the McDonnell Genome Institute at Washington University for library preparation and sequencing. Single-cell suspension samples were processed into nuclei according to 10X Multiome ATAC + Gene Expression (GEX) protocol (CGOOO338). Briefly, cell samples were collected and washed with PBS (with 0.04% BSA), lysed with chilled Lysis Buffer for 4 min, washed 3 times with wash buffer, and resuspended with 10X nuclei buffer at 3000-5000 nuclei/μl. Nuclei samples were subsequently processed using the Chromium 10X genomics instrument, with a target cell number of 7000-10000. The 10X Single Cell Multiome ATAC + Gene Expression v1 kit was used according to the manufacturer's instructions for library preparations. Sequencing of the library was performed using the NovaSeq 6000 System (Illumina).

### Multiomic sequencing analysis workflow

Multiomic sequence files were processed for demultiplexing and analyzed using Cell Ranger ARC v2.0. Genes were mapped and referenced using GRCh38. RStudio was used to perform subsequent analyses. Datasets were further analyzed using Seurat v4 and Signac v1.9.0. For ATAC data, genomic positions were mapped and annotated with EnsDb.Hsapeins.v86 and hg38. Low quality cells were filtered by removing cells with low RNA counts (nCount_RNA < 1000) and low ATAC counts (nCount_ATAC < 1,000); high RNA counts (ranging > 40,000–50,000) and high ATAC counts (ranging > 40,000–50,000); nucleosome signal > 1.25, and TSS enrichment < 2. This resulted in 23,669 cells between 2 samples with a mean 11,485 counts/cell across 3875 genes and 11,769 peak counts/cell across 9,504 genomic regions (WT: 12,930 cells, PRO: 10,739 cells). Gene expression data and ATAC data were processed using SCT transform with cell cycle regression for the former and RunTFIDF and RunSVD for the latter. Dataset integration was performed by anchoring using SCT normalized data and integrative gene expression and ATAC dimension reductions were carried out using PCA and LSI with 12 and 16 components, respectively. Promoter accessibilities were determined from ATAC information using GeneActivity by considering 2000 base pairs upstream of the transcription start site. ATAC peaks were called using MACS2 and linked using LinkPeaks and Cicero to determine cis-regulatory elements. Motif activity was computed by using the chromVAR package and JASPAR version 2020 database.

### Tumor dissociation and sequencing

Proneural tumors were dissociated using the gentleMACS Octo Dissociator and Brain tumor dissociation kit (Miltenyi). Myelin and red blood cells were removed using the MACS Myelin removal beads II and ACK Red Blood Cell lysis buffer, respectively. Dead cells were removed (MACS Dead cell removal microbeads) and viable cells were counted by Trypan blue exclusion. Dissociated tumors cells from 3 mice were processed using the 10X Genomics Chromium Controller and the Chromium Single Cell 3′ v3 Library & Gel Bead Kit following the manufacturer's protocols (https://tinyurl.com/ybpg2pfz). Cells were sequenced at the Washington University School of Medicine Genome Technology Access Center on the NovaSeq 6000 (Illumina) at 2x150bp.

### PRO eNSC tumor scRNA-seq analysis

Alignment and transcript quantification were performed using the Cell Ranger pipeline (10X Genomics, default settings, version 3.0.1). Using the Seurat package in R, cells containing fewer than 200 expressed genes, more than 2000 genes, or containing more than 10% mitochondrial transcripts were removed. Genes that were expressed in fewer than three cells were also removed, resulting in 23,855 cells from 3 mice with a mean of 1602 counts/cell across 834 genes (Mouse-1: 9709 cells, Mouse-2: 8046 cells, Mouse-3: 6100 cells). For each cell, the expression of each gene was normalized to the sequencing depth of the cell, scaled to a constant depth (1 × 106), and log-transformed. The 2000 most variable genes were selected using the variance stabilizing transformation provided in Seurat. Principal component analysis was performed on the variable genes, and 20 components were retained for downstream analyses, including UMAP visualization and unsupervised graph-based clustering (clustering resolution = 0.4). The cell cycle phase was determined using the methodology provided in Seurat, based on the relative expression of phase-specific genes, and cell-cycle signal was removed via multiple regression. Pseudobulk TCGA subtype scoring was performed using the average expression of each gene across all cells and enrichment analysis of gene sets defining Proneural, Classical, and Mesenchymal expression signatures[3]. Additionally, enrichment analysis of single cell TCGA signatures was performed using the *AddModuleScore* function in Seurat and subtype-defining gene sets. Single cells were also analyzed for cell state signatures defined in Neftel et al.[10] using the *AddModuleScore* function and heatmaps were generated using pheatmap v1.0.12. RNA content and velocity analyses were performed as above. Scoring cells by functional annotations defined in Garofano et al.[27] was performed using gene lists kindly provided by the author and *ssMwwGST* function defined in the study. Stacked bar plots were made using dittoSeq v1.6.

### Lentiviral RNAi transduction

The following RNAi target sequences were used for knockdown of genes of interest.

 INSM1.1: GCACGAGAAGCACAAGTACTT
 INSM1i.2: CGGAGAGTCGTTCGCCAGCAA
 TCF4i.1: CACGAAATCTTCGGAGGACAA
 TCF4i.2: CACGAAATCTTCGGAGGACAA
 SOX4i.1: GAAGAAGGTGAAGCGCGTCTA
 SOX4i.2: AGCGACAAGATCCCTTTCATT

For knockdown of INSM1 in B67, B165, and B188 primary GSCs, we utilized GIPZ RNAi which yielded more efficient inhibition.

 INSM1i.A: TCAACATATACATTTAGAG
 INSM1i.B: TGAAGCTTTCTATAAATAG

### PRO eNSC RNAi scRNA-seq preparation and analysis

Multiplexed sequencing for INSM1, TCF4, and SOX4 inhibition in PRO eNSCs was performed using HTO multiplexing as described above and 10X Genomics Chromium Single Cell 3′ v3 Library construction and sequencing as described above. Alignment and transcript quantification were performed using the Cell Ranger pipeline (10X Genomics, default settings, version 3.0.1). Demultiplexing was performed with the Seurat package in R, cells were scored as doublets using consensus calls between hybrid v2.0, scDblFinder v1.12.0, and DoubletFinder v2.0, and outlier analysis performed for counts and features outside 5 standard deviations from their respective means. Doublets, outliers, and genes expressed in fewer than three cells were removed, resulting in 19,098 cells from 4 conditions with a mean of 25,719 counts/cell across 5949 genes (shCtrl: 6766 cells, INSM1i: 3097 cells, TC4i: 4353, SOX4i: 4882 cells). For each cell, the expression of each gene was normalized to the sequencing depth of the cell, scaled to a constant depth (1 × 106), and log-transformed. The 2000 most variable genes were selected using the variance stabilizing transformation provided in Seurat. Principal component analysis was performed on the variable

genes, and 15 components were retained for downstream analyses, including UMAP visualization. The cell cycle phase was determined using the methodology provided in Seurat, based on the relative expression of phase-specific genes, and cell-cycle signal was removed via multiple regression. Spearman similarity comparisons were performed using the average expression of the 250 most variable features across all conditions and performing a correlation analysis using the *cor* function in R. Differential gene expression testing was performed suing the *FindAllMarkers* function in Seurat with logfc.threshold = 0.75 and min.pct = 0.5. Functional scoring was performed as above.

## Analysis of INSM1-dependent IPC-like state in GBM

The scRNA-seq atlas of human embryonic cortical progenitor cells[34] was downloaded and analyzed using the Seurat package in R. High-dimensional Weight Gene Co-expression Network Analysis was performed using hdWGCNA v0.4.03 in R. INSM1 gene signature scoring was performed using the *AddModuleScore* function in Seurat. Functional subtyping was performed as described above, and reference label transfer was accomplished in Seurat using the *FindIntegrationAnchors* and *IntegrateData* functions.

## Statistics and Reproducibility

Sample size is based on effect sizes from prior publications. Unless otherwise stated, experiments were carried out three or more times in three biologically independent samples. Data were reproducible. Samples, cells, and mice used in all experiments were randomized. For all experiments, investigators were blinded as to group allocation during data collection and analysis. For all mouse experiments, experiments were double-blinded--in regard to condition upon cell injection into mice and in regard to endpoints of live bioluminescence and neurological deficit-free survival.

## Reporting summary

Further information on research design is available in the Nature Portfolio Reporting Summary linked to this article.

## Data availability

Single-cell-sequencing datasets generated for this study have been uploaded to the NCBI GEO repository with accession number GSE229901. The preprocessed TCGA-GBM data was downloaded from the following publicly available publication, doi:10.1038/ng.3781 (Barthel et al.)[47]. Additional source data are provided with this paper. The Neftel et al. (2019) 10X and Smart-seq2 data[10] were downloaded from the following publicly accessible site: https://singlecell.broadinstitute.org/single_cell/study/SCP393/single-cell-rna-seq-of-adult-and-pediatric-glioblastoma. The Delgado et al. 10X data[34] were downloaded from the Gene Expression Omnibus with accession number GSE187875. The remaining data are available within the Article, Supplementary Information, or Source Data file. Source data are provided with this paper.

## Code availability

No original code was developed for the purposes of this study. Previously published algorithms and computational tools were utilized as described above.

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

## Acknowledgements

This work was supported by National Institutes of Health grants R01 NS094670, R01 NS106612, R01 NS128470 (to A.H.K.), Hope Center for Neurological Disorders Pilot Grant (to A.H.K. and J.R.M), S10OD027042, P30 CA091842 (to the Molecular Imaging Center, Washington University), S10OD021629-01A1 (to Washington University Center for Cellular Imaging), and T32 DK007120 (to M.I.); the Alvin J. Siteman Cancer Center Siteman Investment Program through funding from The Foundation for Barnes-Jewish Hospital (to A.H.K.); The Christopher Davidson and Knight Family Fund (to A.H.K.); the Duesenberg Research Fund (to A.H.K.); the Edward J. Mallinckrodt Foundation (to J.R.M.); startup funds from the Washington University School of Medicine Department of Medicine (to J.R.M.); Rita Levi-Montalcini Postdoctoral Fellowship from the Washington University School of Medicine Center of Regenerative Medicine (to M.I.); National Institutes of Health (R25 NS090978 to R.H.H.); American Cancer Society (PF-21-149-01-CDP to R.H.H.). We thank members of the A.H.K. and J.R.M. labs for helpful discussions. We would like to thank Drs. Yi-Hsien Chen, Yong Miao, Vijayalingam Selvamani, and Xiaoxia Cui (Washington University School of Medicine Genome Engineering and Stem Cell Center) who performed the CRISPR/Cas9 screening experiments for mutant hESC generation. The MRI studies presented in this work were carried out in the Small Animal MR Facility of the Mallinckrodt Institute of Radiology at Washington University supported by S10OD026913.

## Author contributions

P.D., M.I., C.M., X.Q., D.A., J.R.M., and A.H.K. designed the project and experiments, as well as performed in vitro experiments. P.D., D.D.M., R.V., Y.F., A.V., P.A., B.S., J.K., and M.B.H. performed additional in vitro experiments. P.D., X.Q., R.C., R.H., and T.W.(Timothy Woodiwiss), performed all in vivo experiments. P.D., M.I., C.M., and X.Q. performed all computational analyses and associated cell culture. P.D., J.R.M., and A.H.K. wrote the manuscript. M.I., C.M., H.C., S.P., H.Y., T.W. (Ting Wang), L.F.Z.B., and J.R.M. edited the manuscript. All authors revised and approved the manuscript.

## Competing interests

A.H.K. is a consultant for Monteris Medical and has received research grants from Stryker for a clinical outcomes study about a dural substitute, which have no direct relation to this study. J.R.M. was an employee of and has stock in Sana Biotechnology. M.I. has stock in Vertex Pharmaceuticals. The remaining authors declare no competing interests.

## Additional information

---

¹Taylor Family Department of Neurosurgery, Washington University School of Medicine, St. Louis, MO, USA. ²Department of Genetics, Washington University School of Medicine, St. Louis, MO, USA. ³Division of Endocrinology, Metabolism and Lipid Research, Department of Medicine, Washington University School of Medicine, St. Louis, MO, USA. ⁴Center for Genome Sciences and Systems Biology, Washington University School of Medicine, St. Louis, MO, USA. ⁵Department of Medicine, Washington University School of Medicine, St. Louis, MO, USA. ⁶Division of Hematology, Washington University School of Medicine, St. Louis, MO, USA. ⁷Center for Genome Integrity, Siteman Cancer Center, Washington University School of Medicine, St. Louis, MO, USA. ⁸Sanofi R&D, Vitry-sur-Seine, France. ⁹Department of Biomedical Engineering, Washington University in St. Louis, St. Louis, MO, USA. ¹⁰Department of Immunology, Faculty of Medicine Siriraj Hospital, Mahidol University, Bangkok, Thailand. ¹¹Department of Neurological Surgery, University of Iowa Carver College of Medicine, Iowa City, Iowa, USA. ¹²Department of Neuroscience, Washington University School of Medicine, St. Louis, MO, USA. ¹³Department of Neurology, Washington University school of Medicine, St. Louis, MO, USA. ¹⁴The Brain Tumor Center, Siteman Cancer Center, Washington University School of Medicine, St. Louis, MO, USA. ¹⁵Department of Computer Science and Engineering, Washington University in St. Louis, St. Louis, MO, USA. ¹⁶These authors contributed equally: Matthew Ishahak, Xuan Qu, Colin McCornack. ✉e-mail: jmillman@wustl.edu; alberthkim@wustl.edu

