## [Transparent Peer Review file · Nature Communications]

INSM1 governs a neuronal progenitor state that drives glioblastoma in a human stem cell model

Corresponding Author: Dr Albert Kim

Version 0:

Reviewer comments:

Reviewer #1

(Remarks to the Author)

De Souza and colleagues describe a new model of GBM whereby they begin with human embryonic stem cells, and by Crispr introduce an activating mutation in *tert* and DN mutation in *p53*. Next, they differentiate the resultant cells into NSCs as determined by transition of Oct4 expression into PAX6 expression. Finally, they use Lentivirus delivery to introduce an activated PDGF receptor mutant.

1) The design and execution of these experiments is rigorous and the resultant wt, *tert* mutant, *tert* + *p53* mutant, and *tert*+*p53*+ mutPDGR lines are used for the study.

In summary only the triple mutant cells make GBM-like tumors in orthotopically injected immunocompromised mice (Fig. 1). These resultant tumors are used for all subsequent studies.

2) Generation of at least one additional independently derived tumor forming line would have provided highly beneficial strength to the reproducibility of the outcomes.

3) Why were the *tert* and *p53* mutations introduced into the embryonic stem cells rather than to the neural stem cells as would be the way things happened in normal tumor evolution? Is it possible that by introducing these significant mutations in the more primitive embryonic stem cells might alter how they differentiate and thereby affect the fidelity of the resultant cells?

The validation of the mutations (Fig.2) leads to an interesting and significant observation that *p53* loss results in enhanced *tert* expression. This is experimentally followed up by knock down of wt *p53* which shows similar results in support of the interpretation that it is loss of *p53* function that causes this and not any *p53* gain of function effects.

4) Unfortunately, this mechanism is not pursued further.

Next, multiomics analyses are performed comparing the wt, *tert* mutants, *tert*+*p53* mutant and triple mutant lines. From this point forward, this manuscript is highly computational and only one functional experiment (which will be discussed below) is performed. This reviewer is not a computational biologist so I cannot comment on the rigor of the many analyses. However, taken at face value each analysis leads to an inference about cell states that is never submitted to functional validation.

Italics text is extracted directly from the manuscript results section.

The multiomic analyses somehow point to an "Inferred ordering of PRO eNSC tumor clusters by RNA content." And to a set of genes: "Genes most correlated with the terminal cluster identified by RNA content transcriptional covariance." Further inference from RNA velocity plots that to me appear to have arrows going in many directions, draw the authors' attention to cluster 4 (Fig 4) including INSM1, CD24, STMN2 and Sox 4. Following RNA velocity to "infer" trajectory and "probabilistic fate mapping" in cluster 4, they come up with three genes of interest: TCF4, SOX4, and STMN2.

What next follows is the only functional experiment on the tumor line wherein they efficiently (>75%) knock down the three

genes: To test their functional significance, we first knocked down each of these TFs in PRO eNSCs using two independent lentiviral RNAi each (Fig. 5A, Extended Data Fig. 6A). We observed a significant decrease in self-renewal capacity by extreme limiting dilution following inhibition of each TF, indicating attenuation of malignant transformation in vitro and the likely loss of tumorigenic potential (Fig. 5B, Extended Data Fig. 6B).

5) What they call a stem cell frequency assay to me appears to be just a low cell concentration limiting cell dilution assay which shows that knockdown of each of the genes (TCF4, SOX4, and STMN2) reduces cell number which they call "stem cell frequency". An alternative explanation is that knocking down of any of these three genes just makes the cells sick. Some important controls such as KD of the wt, tert mutants, tert+p53 mutant, and at least one independent triple mutant line would be a minimal requirement.

6) There is incremental buildup of inference upon probability in the figures beyond figure 2 for which there is no systematic functional validation. Omic analyses must be backed by biological testing and validation.

In summary, barring figures 1 & 2, this is a largely computational study that very likely is of state of the art quality. However, there is little biology and no rigorous functional validation of any of the ideas proposed or discussed. This non-functional approach is cropping up in many studies in the field. The most interesting result is the observation that p53 loss may regulate tert expression which is not followed up. Should functional validation be achieved, the use of more than a single cell line should be studied to provide experimental evidence that the results are robust.

Reviewer #2

(Remarks to the Author)

This manuscript is a comprehensive study that covers in vitro CRISPR/Cas9-engineered cells, in vivo tumors from orthotopic implantation, and GBM patient samples. The RNA velocity analysis is a key tool to discover shifts in the transcriptome during cancer evolution by single-cell RNA sequencing rather than acquiring multiple samples in a time-series fashion. The predicted trends are further validated in in vivo orthotopic tumor models and GBM patient samples. By analyzing these transcriptomic shifts, the author successfully identified the transition from stress-related metabolic changes and transitioned toward neuronal progenitor networks driven by transcription factor INSM1.

The manuscript is well-written. The arguments presented within the manuscript are persuasive and well-supported. The RNA velocity analysis is particularly impressive and comprehensive, which covers in vitro, in vivo, and patient sample studies. We have one major comment regarding the model design that warrants additional attention. Additionally, we have provided some comments on the organization of the writing to further strengthen the overall argument.

Major:

1. The Hockemeyer lab spent several years trying to engineer TERTp mutant cells but conventional homology directed repair (HDR) CRISPR-based genome editing was not successful. Instead, they came up with a two-step strategy where the entire promoter was excised and replaced with a mutant promoter (PMID 26194807 and 28818973, the latter already cited in the introduction). The Furnari lab went down the same road several years later and also concluded that conventional CRISPR was unable to generate TERTp mutant clones and ultimately also followed the Hockemeyer strategy (PMID 35325218, already cited in the introduction). This is – to this reviewer's knowledge – the first report of a TERTp mutation engineered using conventional one-step CRISPR. What is unique about the approach that the authors took here that differentiates this from prior attempts in the literature (eg. as cited here)? Why were they successful where others were not? Can the authors include additional validation experiments to QC their TERTp mutant cells in the manuscript supplement? What was their HDR efficiency? How did they select for mutant cells? Is it the mutant allele that's being expressed? The methods describing the knock-in experiment can also be more elaborate, an abbreviated description would only be appropriate if a prior publication can be cited.

Minor:

1. Fig. 2: The title "TP53 loss-of-function increases TERT expression in GBM" sounds like an exciting new finding. However, this argument lacks follow-up data within this manuscript, making it less relevant to the main topic. I assume that the purpose of Fig. 2 is to validate the TERT promoter mutation in CRISPR-edited eNSCs. If this is the case, Fig. 2A-F is sufficient for functional validation of the TERT promoter mutation. Consider moving Fig. 2F-I to the supplementary data.

2. Line 150 - 165: Similar to the point above, consider merging this paragraph with the previous one as an additional finding or move it to the discussion section.

3. Line 168 - 174: There's no one-size-fits-all method. It's understood that PCA sometimes doesn't give the expected results. You could simply state that PCA didn't work and you then tried the CytoTRACE algorithm, saving the explanation for the discussion section. This may improve the flow of reading.

4. Fig. 3B: How about pathway analysis in other eNSCs with single and double mutations? Does this phenomenon exclusively occur in the PRO eNSCs?

5. Fig. 5B: Similar to the previous point, does this phenomenon exclusively occur in the PRO eNSCs rather than in WT, single, or double mutants?

6. Fig. 5: I couldn't find any data related to "metabolism" in Fig. 5, even though the title is "INSM1 inhibition in PRO eNSCs induces metabolic and developmental regression." Consider putting Extended Data Fig. 7E to Fig. 5 to improve the

argument.

7. Fig. 6G: Cluster 6 and Cluster 13 also have a high score of INSM1 signature. What pathways and genes are involved in these clusters? Is there any discussion about them?
8. Fig. 5H: It should be INSM1i on the right side of the y-axis label.
9. Line 648: Do you mean PRO hGBM?
10. Line 402: It should be Fig. 6F
11. Chromatin immunoprecipitation in the method section: Did you miss describing the conjugation of antibodies to the beads prior to overnight incubation?
12. Extended Data Fig. 8: The last x-axis label is off.
13. Duplicated figures after line 1254.

Reviewer #3

(Remarks to the Author)

Reviewer #4

(Remarks to the Author)

In the submitted manuscript, the authors provide a human framework to introduce oncogenic drivers and mutations in specific neural cell lineages in a step-wise fashion. This is accompanied by detailed transcriptional and epigenetic characterization of tumor-associated cell states tied to specific and combinations of oncogenic drivers. In glioblastoma, this has relevance given the cellular diversity of transcriptionally distinct populations, and their significant role in disease progression and therapeutic resistance. From a functional perspective, the authors demonstrate the disruption of key factors that regulate proneural cell state, specifically INSM1 inhibition, have important effects on cancer stem cell frequency and transcriptional lineage programs, that may have functional impact on GBM tumor development. While the isogenic modeling of GBM and associated cell states is of particular relevance, in addition to studies of transcriptional evolution, several outstanding questions remain to be addressed:

- 1) It's clear from the author's detailed transcriptional analysis that PRO tumor models have cell types that can be classified as proneural, mesenchymal, and classical. However, unclear is how transcriptionally similar are tumors to primary GBM tumors, or proneural signatures from GBM patient samples. Additionally, would DNA methylation classification of PRO eNSC tumors place these models in similarity with primary GBM tumors, and those enriched with proneural signatures.
- 2) The functional impact on INSM1 and stem cell state is not clearly defined. How does INSM1 loss-of-function alter tumor initiation, of the tumors (assuming they arise) now enriched in other GBM cell states that are capable of driving tumor development.
- 3) The importance of INSM1 and its effects on proneural state need to be validated in patient derived tumor models. A related concern is whether cell lineage programs modeled are reflective of neural stem cell populations seen in the adult setting, where glioblastoma multiforme is more frequently observed.

Version 1:

Reviewer comments:

Reviewer #2

(Remarks to the Author)

The authors have addressed all my concerns. The additional pathway analysis and Seahorse XF analysis support the hypothesis that gene expression shifts from promoting metabolism to neurodevelopment during tumor evolution.

- FPB

Reviewer #3

(Remarks to the Author)

I appreciate the effort the authors have made to address the concerns I previously raised. I am pleased to see that most of my concerns have been adequately addressed, including clarifications in figure legends, additional experimental validations, and improvements in data presentation. The inclusion of in vivo experiments to validate the role of INSM1 further strengthens the impact of this study.

However, I still have concerns regarding the scRNA-seq analysis presented in Figure 3. While I understand that the dataset was processed with roughly similar cell numbers across experimental groups and that RNA velocity was used as an orthogonal validation, my concern remains regarding the appropriateness of CytoTRACE for this context. Given the fundamental differences between developmental differentiation and cancer cell state transitions, I would strongly encourage a re-analysis of the scRNA-seq dataset using an alternative clustering approach—such as unbiased clustering based on

transcriptomic similarity across pooled cells from all conditions. This would help confirm whether the observed differentiation hierarchy remains consistent across methods.

Overall, I find the manuscript significantly improved and believe this additional analysis will further strengthen the conclusions.

Reviewer #4

(Remarks to the Author)

In the revised manuscript, the authors have responded with new in vitro, in silico, and in vivo data and addressed my initial comments thoroughly. I have no further or new comments.

Reviewer #5

(Remarks to the Author)

Reviewer #6

(Remarks to the Author)

Responsiveness to Reviewer A's comments:

The authors are highly responsive to. The authors adequately addressed most of the comments by reviewer A with new supporting data and satisfactorily responded. However, the response to comment #6 "There is incremental buildup of inference upon probability in the figures beyond figure 2 for which there is no systematic functional validation. Omic analyses must be backed by biological testing and validation" is insufficient.

New Comments:

1. PRO (TERTp C228T, TP53 G473A R248Q, PDGFRAD284V) model does not recapitulate the actual PDGFRA mutation in clinical adult GBM tumors. The PDGFRAD284V mutation occurs in pediatric high-grade glioma (pHGG) in children and often associated with H3 K27M mutated pHGG. In adult GBM, PDGFRAD284V mutation is rare or highly infrequent. In contrast, a PDGFRA $\Delta 7-8$ mutation has been reported and characterized in adult GBM tumors. Thus, the PRO eNSC GBM model presented in this study does not recapitulate clinical presentation of PDGFRA mutation in adult GBM.
2. The advance of development and characterization of the PRO eNSC GBM model system in relation to recently reported human induced pluripotent stem cell (hiPSC)-derived glioma avatar models is not clear. The rationale of using human embryonic stem cells versus neural stem cell as the starting cell model is not strong. The response by the authors to this comment is insufficient.
3. The rationale of selection of INSM1 as the key transcription factor downstream of PRO mutation drivers is weak. As shown in Figure 6D, in relation to SOX4 and TCF4, the actual level of INSM1 in human GBM is much lower. The authors should also present the relative expression of other TFs in 6D, STMN2, DCX, SOX4, TCF4 and CD24 in the form of 6B and justify the focus of INSM1 in this study.
4. The cellular function of INSM1 is transcriptional repressor in neurogenesis and neuroendocrine cell differentiation during embryonic and/or fetal development. INSM1 represses gene transcription by recruiting chromatin-modifying factors, such as HDAC1, HDAC2, KDM1, and RCOR1 histone deacetylases. However, the functional and consequential studies by knockdown of INSM1 (INSMi) in PRO models presented here failed to relate to the basic cellular function of INSM1.

Version 2:

Reviewer comments:

Reviewer #3

(Remarks to the Author)

I appreciate the authors' efforts to address my concerns regarding the scRNA-seq analysis. The inclusion of RNA velocity as an orthogonal validation is a valuable addition, and the roughly similar cell numbers across experimental groups partially mitigate the sampling bias concerns. However, a few issues remain:

CytoTRACE Suitability: While RNA velocity supports the trajectory findings, the fundamental concern about CytoTRACE's applicability to cancer contexts—where "differentiation" differs from developmental processes—remains unaddressed. The authors should acknowledge this limitation or test an alternative trajectory method (e.g., Monocle 3 or Slingshot) to confirm their results.

Alternative Analysis: The authors did not perform the suggested downsampling to equal cell numbers and re-clustering, which would have directly tested the robustness of their conclusions. Although their existing pooled clustering approach is reasonable, this exact reanalysis would strengthen confidence in the results.

Given the overall strength of the study and the authors' good-faith response, I recommend minor revisions to address these points. Specifically, the authors should:

Discuss the limitations of applying CytoTRACE to cancer data.
Consider performing the downsampling and re-clustering analysis or provide a stronger justification for why it is unnecessary.
These revisions will enhance the manuscript's rigor without requiring extensive new experiments.

Reviewer #6

(Remarks to the Author)

In the revised manuscript, the authors have addressed my comments with additional data as well text revisions. No further concerns.

Point-By-Point Response to Reviewers' Comments

Manuscript#: NCOMMS-23-36248-T

Title: An aberrant INSM1-dependent intermediate neuronal progenitor state drives tumorigenesis in a human stem cell model of glioblastoma

We appreciate the reviewers' insightful comments and suggestions, which have been invaluable in enhancing the quality of our revised manuscript. We carefully considered each comment and have made substantive revisions to address the concerns raised during the review process. Below, we provide our detailed response to each comment and description of related revisions in our manuscript. We are pleased to say that we have addressed all of the reviewers' comments, and as a result our manuscript is significantly improved.

Reviewer #1

De Souza and colleagues describe a new model of GBM whereby they begin with human embryonic stem cells, and by Crispr introduce an activating mutation in tert and DN mutation in p53. Next, they differentiate the resultant cells into NSCs as determined by transition of Oct4 expression into PAX6 expression. Finally, they use Lentivirus delivery to introduce an activated PDGF receptor mutant.

1) The design and execution of these experiments is rigorous and the resultant wt, tert mutant, tert + p53 mutant, and tert+ p53+ mutPDGR lines are used for the study.

In summary only the triple mutant cells make GBM-like tumors in orthotopically injected immunocompromised mice (Fig. 1). These resultant tumors are used for all subsequent studies.

2) Generation of at least one additional independently derived tumor forming line would have provided highly beneficial strength to the reproducibility of the outcomes.

3) Why were the tert and p53 mutations introduced into the embryonic stem cells rather than to the neural stem cells as would be the way things happened in normal tumor evolution? Is it possible that by introducing these significant mutations in the more primitive embryonic stem cells might alter how they differentiate and thereby affect the fidelity of the resultant cells?

The validation of the mutations (Fig.2) leads to an interesting and significant observation that p53 loss results in enhanced tert expression. This is experimentally followed up by knock down of wt p53 which shows similar results in support of the interpretation that it is loss of p53 function that causes this and not any p53 gain of function effects.

4) Unfortunately, this mechanism is not pursued further.

Next, multiomics analyses are performed comparing the wt, tert mutants, tert+p53 mutant and triple mutant lines. From this point forward, this manuscript is highly computational and only one functional experiment (which will be discussed below) is performed. This reviewer is not a computational biologist so I cannot comment on the rigor of the many analyses. However, taken at face value each analysis leads to an inference about cell states that is never submitted to functional validation.

Italics text is extracted directly from the manuscript results section.

The multiomic analyses somehow point to an "Inferred ordering of PRO eNSC tumor clusters by RNA content." And to a set of genes: "Genes most correlated with the terminal cluster identified by RNA content transcriptional covariance." Further inference from RNA velocity plots that to me appear to have arrows going in many directions, draw the authors' attention to cluster 4 (Fig 4) including INSM1, CD24, STMN2

and Sox 4. Following RNA velocity to “infer” trajectory and “probabilistic fate mapping” in cluster 4, they come up with three genes of interest: TCF4, SOX4, and STMN2.

What next follows is the only functional experiment on the tumor line wherein they efficiently (>75%) knock down the three genes: To test their functional significance, we first knocked down each of these TFs in PRO eNSCs using two independent lentiviral RNAi each (Fig. 5A, Extended Data Fig. 6A). We observed a significant decrease in self-renewal capacity by extreme limiting dilution following inhibition of each TF, indicating attenuation of malignant transformation in vitro and the likely loss of tumorigenic potential (Fig. 5B, Extended Data Fig. 6B).

5) What they call a stem cell frequency assay to me appears to be just a low cell concentration limiting cell dilution assay which shows that knockdown of each of the genes (TCF4, SOX4, and STMN2) reduces cell number which they call “stem cell frequency”. An alternative explanation is that knocking down any of these three genes just makes the cells sick. Some important controls such as KD of the wt, tert mutants, tert+p53 mutant, and at least one independent triple mutant line would be a minimal requirement. 6) There is incremental buildup of inference upon probability in the figures beyond figure 2 for which there is no systematic functional validation. Omic analyses must be backed by biological testing and validation.

In summary, barring figures 1 & 2, this is a largely computational study that very likely is of state of the art quality. However, there is little biology and no rigorous functional validation of any of the ideas proposed or discussed. This non-functional approach is cropping up in many studies in the field. The most interesting result is the observation that p53 loss may regulate tert expression which is not followed up. Should functional validation be achieved, the use of more than a single cell line should be studied to provide experimental evidence that the results are robust.

1. The design and execution of these experiments is rigorous and the resultant wt, tert mutant, tert + p53 mutant, and tert+ p53+ mutPDGR lines are used for the study.

We thank the reviewer for these positive comments regarding the rigor and design of our novel human stem cell model of GBM.

2. Generation of at least one additional independently derived tumor forming line would have provided highly beneficial strength to the reproducibility of the outcomes.

We agree with this point. In fact, the data in the original Fig. 1H represent 2 independent clones per genetic background, shown as aggregated experimental groups. To better emphasize this point, the Kaplan-Meier survival curve in Fig. 1H has now been modified to highlight the use of 2 clones per genotype and the reproducibility of the phenotypic data. In addition, we have added a new Supplementary Fig. 2 to separate out the survival curves of the individual clones and highlight the consistent tumorigenic phenotype of the second triple mutant PRO clone by survival outcomes, MRI, and histopathology.

Supplementary Fig. 2: PRO eNSCs form tumors in mice that recapitulate GBM. **A** Kaplan-Meier survival curves for each genotype of clone 1. **B** Kaplan-Meier survival curves for each genotype of clone 2. **C** T2-weighted MRI images of eNSC-xenografted mice using clone 2 for each genotype. Red outlines indicate pathological lesions in the coronal plane of injection site (scale bar = 1 mm; L = left, P = posterior, R = right). **D** Immunofluorescence imaging of a PRO eNSC brain tumor using clone 2. Frozen sections were stained for Human Nuclear Antigen (488 nm) and DAPI (scale bar = 0.5 mm). Inset images display magnified fields of view and merged images (scale bar = 50 μ m). **E** H&E staining of a PRO eNSC brain tumor using clone 2 demonstrates characteristic histopathology of GBM. Pink necrotic center (n) surrounded by pseudopalisading tumor cells (arrowheads), (left, scale bar = 200 μ m). Tumor hypercellularity and mitotic atypia (right, scale bar = 50 μ m).

3. Why were the *tert* and *p53* mutations introduced into the embryonic stem cells rather than to the neural stem cells as would be the way things happened in normal tumor evolution? Is it possible that by introducing these significant mutations in the more primitive embryonic stem cells might alter how they differentiate and thereby affect the fidelity of the resultant cells.

Given the finite replicative and experimentally useful lifespan of neural stem cells (NSC), which are prone to differentiation and terminal senescence with increasing passage number, we introduced the *TERT* and *TP53* mutations in the indefinitely replicating embryonic stem cells. This minimized the number of NSC passages in culture for generating and validating mutation status and maximized experimental use before later passage numbers (>20). As shown in Fig. 1C and the original Supplementary Fig. 1 qPCR data, the engineered NSCs for each initial genotype demonstrated no significant differences in the expression of cell state markers compared to the wildtype NSCs, indicating little to no effect of genetic background on NSC differentiation status. We have verified this through *new quantification of immunofluorescence of each cell*

state marker between genotypes (*new Supplementary Fig. 1C*) and have added new qPCR data for NSC marker *PAX6* to demonstrate that expression of this key cell state marker is not altered by addition of PDGFRA mutant to double mutant NSCs (*Supplementary Fig. 1E, right-most graph*).

Supplementary Fig. 1: Engineered NSCs display mutant-specific signaling and malignant transformation in vitro. **A** Table of next generation sequencing (NGS) read counts at *TERT* promoter and *TP53* exon 6 CRISPR-mutated sites in two sets of wildtype, *TERT*p, and *TERT*p+*TP53* hESCs. **B** Immunocytochemistry (ICC) for stem cell markers OCT4 and PAX6 (488 nm), as well as DAPI, in hESC clone set 2 and their differentiation-matched NSCs (scale bar = 50 μ M). **C** Quantification of ICC fluorescence intensity normalized to cell number ($n=3$, ANOVA, ns). **D** Quantitative RT-PCR for analysis of OCT4 mRNA expression normalized to ACTIN and GAPDH expression in two independent genetic clone sets of mutant hESC and matched NSC clones ($n=3$, ANOVA, *** $P<0.001$, **** $P<0.0001$). **E** Quantitative RT-PCR for analysis of *PAX6* mRNA expression normalized to ACTIN and GAPDH expression in two clone sets of hESC and matched NSC clones ($n=3$, except for right-most histogram where $n=2$ per clone, ANOVA, * $P<0.05$, ** $P<0.01$, *** $P<0.001$). **F** Immunoblot analysis for mutant *PDGFRA* expression and activity in eNSCs with indicated antibodies. Representative of three independent replicates. **G** Flow cytometric analysis of EdU incorporation during eNSC *in vitro* culture ($n=5$, ANOVA, ns). **H** Extreme limiting dilution assay of serial mutant eNSC genotypes. Data represent mean \pm SEM ($n=3$, ANOVA, ** $P<0.01$, **** $P<0.0001$). **I** Representative soft agar colony formation assay of eNSCs (left) and

quantification (right). Data represent mean \pm SEM ($n=3$, ANOVA, $**P<0.01$, $***P<0.001$, $****P<0.0001$).

4. The validation of the mutations (Fig. 2) leads to an interesting and significant observation that p53 loss results in enhanced *tert* expression. This is experimentally followed up by knock down of wt p53 which shows similar results in support of the interpretation that it is loss of p53 function that causes this and not any p53 gain of function effects. Unfortunately, this mechanism is not pursued further.

We appreciate the reviewer's positive comments on the significance of the interaction between *TERT* promoter mutation and *TP53* loss of function and are also interested in the mechanism of this intriguing phenomenon. However, based on comments by Reviewers #2 and #3, to sharpen the focus of the paper on the identified neuronal progenitor program, we have decreased the text regarding mutant *TP53* effects on the *TERT* mutant promoter in Results and have moved several Fig. 2 panels to Supplementary Figures. At the same time, we have attempted to provide some preliminary insights as to why *TP53* mutation increases *TERT* expression in a new *Supplementary Fig. 3H*, in which we show that *TP53* mutation in the setting of *TERT* promoter mutation in NSCs increases the expression of GABP β 1, an essential part of the GABP TF complex, compared to *TERT* promoter mutation alone. We briefly refer to these new preliminary data in the Discussion section. We will clearly be studying the mechanism of this significant phenomenon systematically in future studies.

H

Supplementary Fig. 3: *TP53* loss of function increases *TERT* expression in GBM. H Analysis of *TERT* mRNA expression in TCGA-GBM bulk RNA data where samples are stratified by *TP53* mutation status from matched whole exome data (Student's t test, $*P<0.05$).

5. What they call a stem cell frequency assay to me appears to be just a low cell concentration limiting cell dilution assay which shows that knockdown of each of the genes (*TCF4*, *SOX4*, and *STMN2*) reduces cell number which they call "stem cell frequency." An alternative explanation is that knocking down of any of these three genes just makes the cells sick. Some important controls such as KD of the wt, *tert* mutants, *tert*+p53 mutant, and at least one independent triple mutant line would be a minimal requirement.

Based on these comments, we have performed new experiments demonstrating first that knockdown of our major candidate epigenetic driver *INSM1* in a *second* triple mutant *PRO* eNSC clone also decreases self-renewal capacity by extreme limiting dilution assay (new *Supplementary Fig. 10A*) and that *INSM1* knockdown in wildtype eNSCs substantially decreases *INSM1* mRNA levels (new *Supplementary Fig. 10B*) but does not significantly change self-renewal capacity (*Supplementary Fig. 10C*) or decrease glycolytic

rate by Seahorse XF analysis (*new Supplementary Fig. 10E*). These data suggest that inhibition of INSM1 per se is not likely to be required for cellular health.

Supplementary Fig. 10: Inhibiting transcriptional drivers of PRO eNSC evolution disrupts self-renewal capacity and stress-related glycolytic pathways. **A** Self-renewal capacity was measured by extreme limiting dilution assay using PRO eNSCs (clone 2) following INSM1 RNAi (vs. control (shCtrl)) or using PRO eNSCs (clone 1) following TCF4 and SOX4 RNAi (two RNAi per each targeted gene were combined in plot for TCF4i and SOX4i). Data represent mean \pm SEM ($n=3$, ANOVA, * $P<0.05$, ** $P<0.01$). **B** Expression of INSM1 mRNA by qRT-PCR normalized to ACTIN and GAPDH expression in WT eNSCs (clone 1) following INSM1 RNAi. Data represent mean \pm SEM ($n=3$, ANOVA, ** $P<0.01$). **C** Extreme limiting dilution assay using WT eNSCs (clone 1) following INSM1 RNAi. Data represent mean \pm SEM ($n=3$, ANOVA, ns). **D** Seahorse XF analysis of glycolysis-dependent extracellular acidification rate in WT eNSCs (clone 1) following INSM1 RNAi. A slight increase in glycolytic activity was observed with only one INSM1 RNAi. Data represent mean \pm SEM. ($n=8$, ANOVA, **** $P<0.0001$). **E** INSM1 mRNA expression by qRT-PCR normalized to ACTIN and GAPDH expression in B67 primary human GBM cells following *INSM1* RNAi. Data represent mean \pm SEM ($n=3$, ANOVA, ** $P<0.01$).

6) There is incremental buildup of inference upon probability in the figures beyond figure 2 for which there is no systematic functional validation. Omic analyses must be backed by biological testing and validation.

We would agree with the reviewer that functional validation is ultimately needed for any conclusions derived solely from transcription or chromatin-based data, which are the subjects of Fig. 3 and 4, but respectfully disagree with the reviewer's comment that all figures beyond Fig. 2 represent merely inferences or probabilities. Our main goal in Fig. 3 was to determine the cell states and epigenetic modifiers that differed between the four eNSC genotypes *in vitro* given our initial core findings that PRO eNSCs exhibit higher colony formation on soft agar, have higher self-renewal capacity, and produce reliable and robust

infiltrative brain tumors in mouse brain. We would point out that Fig. 3C shows functional metabolic data that confirms some of the differential gene changes we observed in wildtype vs. PRO eNSCs. Our main goal in Fig. 4 was to 1) show that PRO eNSC brain tumors resemble human PRO GBMs at the transcriptional level and 2) identify *in vivo* cell states and specific epigenetic modifiers in PRO brain tumors that converge with our *in vitro* data. In response to the reviewer's request for additional functional validation, we have added *new in vivo experiments to test the role of INSM1 in PRO eNSC tumorigenicity (new Fig. 6E)*. We orthotopically transplanted *INSM1* knockdown PRO eNSCs using two independent RNAi hairpins (vs. control RNAi) into the brains of athymic nude mice. Importantly, mice bearing *INSM1* knockdown PRO eNSCs exhibited significantly increased survival compared to mice bearing control RNAi-infected PRO eNSCs (*new Fig. 6E*). *INSM1* RNAi inhibited brain tumor formation by PRO eNSCs as monitored by MRI (*new Fig. 6F*). These results indicate that *INSM1* is critical for the *in vivo* tumor formation of PRO cells.

Fig. 6: *INSM1* is upregulated in human GBM tumors and is critical for *in vivo* tumorigenicity. Kaplan–Meier curves demonstrated increased survival in animals bearing *INSM1* knockdown PRO eNSCs compared with those bearing control RNAi-infected PRO eNSCs (clone 1 and 2 results combined). Cells were orthotopically implanted in mice (*total n*=15; *n*=10 for PRO clone 1; *n*=5 for PRO clone 2, log-rank, *****P*<0.0001). **Right.** T2-weighted MRI images of mice 200 days post-xenograft with four serial mutant eNSCs (clone 1 set). Red outlines indicate pathological lesions in the coronal plane of the injection site (scale bar = 1 mm; L = left, P = posterior, R = right).

Additionally, in another set of *new experiments*, to confirm the metabolic role of *INSM1* in PRO eNSCs, we have performed functional analyses using Seahorse XF analyses *in vitro*, which revealed a decrease in glycolytic activity following *INSM1* knockdown (*new Fig. 5I, left*). In addition, we knocked down *INSM1* in an orthogonal model system—primary human GBM stem cells, a system with which we have long-term experience (Mao, Gujar, et al, *Cell Rep*, 2015; Gujar et al, *PNAS*, 2016; Mahlokozera et al, *Nat Comm*, 2021). Using a primary GBM stem cell line (B67), we have performed *new* metabolic experiments (*new Fig. 5I, right*), which similarly demonstrated a reduction in glycolytic capacity following *INSM1* knockdown.

Fig. 5I: Seahorse XF analysis of glycolysis-dependent extracellular acidification rate in PRO eNSCs (left) and B67 primary human GBM cells (right) following inhibition of INSM1 using two independent RNAi. Data represent mean \pm SEM (GBM: $n=8$; PRO eNSC: $n=4$, ANOVA, **** $P<0.0001$).

Reviewer #2

This manuscript is a comprehensive study that covers in vitro CRISPR/Cas9-engineered cells, in vivo tumors from orthotopic implantation, and GBM patient samples. The RNA velocity analysis is a key tool to discover shifts in the transcriptome during cancer evolution by single-cell RNA sequencing rather than acquiring multiple samples in a time-series fashion. The predicted trends are further validated in in vivo orthotopic tumor models and GBM patient samples. By analyzing these transcriptomic shifts, the author successfully identified the transition from stress-related metabolic changes and transitioned toward neuronal progenitor networks driven by transcription factor INSM1.

The manuscript is well-written. The arguments presented within the manuscript are persuasive and well-supported. The RNA velocity analysis is particularly impressive and comprehensive, which covers in vitro, in vivo, and patient sample studies. We have one major comments regarding the model design that warrants additional attention. Additionally, we have provided some comments on the organization of the writing to further strengthen the overall argument.

Major:

1. The Hockemeyer lab spent several years trying to engineer TERTp mutant cells but conventional homology directed repair (HDR) CRISPR-based genome editing was not successful. Instead, they came up with a two-step strategy where the entire promoter was excised and replaced with a mutant promoter (PMID 26194807 and 28818973, the latter already cited in the introduction). The Furnari lab went down the same road several years later and also concluded that conventional CRISPR was unable to generate TERTp mutant clones and ultimately also followed the Hockemeyer strategy (PMID 35325218, already cited in the introduction). This is – to this reviewer’s knowledge – the first report of a TERTp mutation engineered using conventional one-step CRISPR. What is unique about the approach that the authors took here that differentiates this from prior attempts in the literature (eg. as cited here)? Why were they successful where others were not? Can the authors include additional validation experiments to QC their TERTp mutant cells in the manuscript supplement? What was their HDR efficiency? How did they select for mutant cells? Is it the mutant allele that’s being expressed? The methods describing the knock-in experiment can also be more elaborate, an abbreviated description would only be appropriate if a prior publication can be cited.

Minor:

1. Fig. 2: The title "TP53 loss-of-function increases TERT expression in GBM" sounds like an exciting new finding. However, this argument lacks follow-up data within this manuscript, making it less relevant to the

main topic. I assume that the purpose of Fig. 2 is to validate the TERT promoter mutation in CRISPR-edited eNSCs. If this is the case, Fig. 2A-F is sufficient for functional validation of the TERT promoter mutation. Consider moving Fig. 2F-I to the supplementary data.

2. Line 150 - 165: Similar to the point above, consider merging this paragraph with the previous one as an additional finding or move it to the discussion section.

3. Line 168 - 174: There's no one-size-fits-all method. It's understood that PCA sometimes doesn't give the expected results. You could simply state that PCA didn't work and you then tried the CytoTRACE algorithm, saving the explanation for the discussion section. This may improve the flow of reading.

4. Fig. 3B: How about pathway analysis in other eNSCs with single and double mutations? Does this phenomenon exclusively occur in the PRO eNSCs?

5. Fig. 5B: Similar to the previous point, does this phenomenon exclusively occur in the PRO eNSCs rather than in WT, single, or double mutants?

6. Fig. 5: I couldn't find any data related to "metabolism" in Fig. 5, even though the title is "INSM1 inhibition in PRO eNSCs induces metabolic and developmental regression." Consider putting Extended Data Fig. 7E to Fig. 5 to improve the argument.

7. Fig. 6G: Cluster 6 and Cluster 13 also have a high score of INSM1 signature. What pathways and genes are involved in these clusters? Is there any discussion about them?

8. Fig. 5H: It should be INSM1i on the right side of the y-axis label.

9. Line 648: Do you mean PRO hGBM?

10. Line 402: It should be Fig. 6F

11. Chromatin immunoprecipitation in the method section: Did you miss describing the conjugation of antibodies to the beads prior to overnight incubation?

12. Extended Data Fig. 8: The last x-axis label is off.

13. Duplicated figures after line 1254.

This manuscript is a comprehensive study that covers in vitro CRISPR/Cas9-engineered cells, in vivo tumors from orthotopic implantation, and GBM patient samples. The RNA velocity analysis is a key tool to discover shifts in the transcriptome during cancer evolution by single-cell RNA sequencing rather than acquiring multiple samples in a time-series fashion. The predicted trends are further validated in in vivo orthotopic tumor models and GBM patient samples. By analyzing these transcriptomic shifts, the author successfully identified the transition from stress-related metabolic changes and transitioned toward neuronal progenitor networks driven by transcription factor INSM1. The manuscript is well-written. The arguments presented within the manuscript are persuasive and well-supported. The RNA velocity analysis is particularly impressive and comprehensive, which covers in vitro, in vivo, and patient sample studies. We have one major comment regarding the model design that warrants additional attention. Additionally, we have provided some comments on the organization of the writing to further strengthen the overall argument.

We thank the reviewer for the overall positive comments on our study and have endeavored to address all of the reviewer's constructive comments.

1. The Hockemeyer lab spent several years trying to engineer TERTp mutant cells but conventional homology directed repair (HDR) CRISPR-based genome editing was not successful. Instead, they came up with a two-step strategy where the entire promoter was excised and replaced with a mutant promoter (PMID 26194807 and 28818973, the latter already cited in the introduction). The Furnari lab went down the same road several years later and also concluded that conventional CRISPR was unable to generate TERTp

mutant clones and ultimately also followed the Hockemeyer strategy (PMID 35325218, already cited in the introduction). This is – to this reviewer’s knowledge – the first report of a TERTp mutation engineered using conventional one-step CRISPR. What is unique about the approach that the authors took here that differentiates this from prior attempts in the literature (eg. as cited here)? Why were they successful where others were not? Can the authors include additional validation experiments to QC their TERTp mutant cells in the manuscript supplement? What was their HDR efficiency? How did they select for mutant cells? Is it the mutant allele that’s being expressed? The methods describing the knock-in experiment can also be more elaborate, an abbreviated description would only be appropriate if a prior publication can be cited.

We apologize for the confusion regarding the generation of our human stem cell model for the *TERT* promoter mutation. We indeed recognized that the *TERT* promoter may be challenging to target given the high GC content of the region. However, we did ultimately use a single-step donor-based strategy to generate the *C228T TERT* promoter mutant clones. We think the key difference is our use of a ribonucleoprotein (RNP) approach rather than transient transfection of Cas9- and guide RNA (gRNA)-expressing plasmids. In the Hockemeyer study, transfection of the plasmids encoding Cas9 and gRNAs was toxic to cells, likely due to ongoing high expression of gRNAs with off-target effects, thereby preventing mutant clone generation using the single-step approach. The RNP approach is known to substantially reduce off-target effects and also lead to more rapid genome editing than a plasmid-based approach. (Kim S, Kim D, Cho SW, Kim J, Kim J-S, Highly efficient RNA-guided genome editing in human cells via delivery of purified Cas9 ribonucleoproteins, *Genome Res* 24 (2014) 1012–1019. Lin S, Staahl B, Alla RK, Doudna JA, Enhanced homology-directed human genome engineering by controlled timing of CRISPR/Cas9 delivery, *Elife* 3 (2014) e04766.)

In brief, in collaboration with the McDonnell Genome Institute Genome Engineering & Stem Cell Core, a synthetic gRNA targeting the promoter sequence and a donor single-stranded oligodeoxynucleotide template harboring the *C228T* mutation for knock-in were purchased (IDT) and complexed with Cas9 recombinant protein. Then Cas9-gRNA RNPs were transfected into H1 ESCs. Transfected cells were single cell sorted into 96-well plates, and clones from single cell colonies were selected and screened using next generation sequencing to analyze the target site region as those harboring knock-in mutation. Mutations were validated by next generation (Supplementary Fig. 1A) and Sanger sequencing (Fig. 1B).

1. Fig. 2: The title "TP53 loss-of-function increases TERT expression in GBM" sounds like an exciting new finding. However, this argument lacks follow-up data within this manuscript, making it less relevant to the main topic. I assume that the purpose of Fig. 2 is to validate the TERT promoter mutation in CRISPR-edited eNSCs. If this is the case, Fig. 2A-F is sufficient for functional validation of the TERT promoter mutation. Consider moving Fig. 2F-I to the supplementary data.

We agree with the reviewer that the finding of increased *TERT* expression with *TP53* loss of function in the context of the *TERT* promoter mutant is very interesting but also agree that this observation is not the focus of the study. We have therefore moved Fig. 2G-I to Supplementary Fig. 3SE-F. We plan to follow up the mechanism underlying the interaction between the *TERT* promoter mutant and *TP53* loss of function in future studies.

2. Line 150 - 165: Similar to the point above, consider merging this paragraph with the previous one as an additional finding or move it to the discussion section.

We have merged the two paragraphs as suggested by the reviewer in the Results section.

3. Line 168 - 174: There's no one-size-fits-all method. It's understood that PCA sometimes doesn't give the expected results. You could simply state that PCA didn't work and you then tried the CytoTRACE algorithm, saving the explanation for the discussion section. This may improve the flow of reading.

As recommended, we have shortened the discussion of the limits of PCA for this dataset in the Results section.

4. Fig. 3B: How about pathway analysis in other eNSCs with single and double mutations? Does this phenomenon exclusively occur in the PRO eNSCs?

Based on these suggestions, we have performed *new pathway analyses* with the genes correlated with the single and double mutant eNSCs in *new Supplementary Fig. 4E* and also present them below.

Interestingly, single and double mutant eNSCs largely displayed features of stress-related metabolism, including protein ubiquitination and turnover, as well as global epigenetic signaling, with less prominent features of neuronal development.

E TERTp and TP53-associated RNA content Gene Ontology

Supplementary Fig. 4: Hashtag oligos allow multiplexed scRNA-seq for trajectory analysis of mutant eNSCs. E Gene ontology of signature representing the middle 50% of inferred ordering associated with *TERTp* and *TERTp+TP53* eNSCs ($n=1,141$). Circle size represents number of genes within each ontology and color indicates log-transformed FDR.

5. Fig. 5B: Similar to the previous point, does this phenomenon exclusively occur in the PRO eNSCs rather than in WT, single, or double mutants?

We have performed new experiments demonstrating first that knockdown of our major candidate epigenetic driver *INSM1* decreases self-renewal capacity in a *second* triple mutant PRO eNSC clone (*new Supplementary Fig. 10A*), suggesting that this is a robust effect in 2 distinct clones. Second, we found that although *INSM1* knockdown in wildtype NSCs substantially decreases *INSM1* mRNA levels (*new Supplementary Fig. 10B*), it does not significantly change self-renewal capacity (*Supplementary Fig. 10C*). We are very interested in additional mutation-specific roles for *INSM1* as well as its regulation in these

different genetic contexts but believe that given the main focus on the tumorigenic PRO eNSCs, these experiments are beyond the scope of the current study.

Supplementary Fig. 10: Inhibiting transcriptional drivers of PRO eNSC evolution disrupts self-renewal capacity and stress-related glycolytic pathways. **A** Self-renewal capacity was measured by extreme limiting dilution assay using PRO eNSCs (clone 2) following INSM1 RNAi (vs. control (shCtrl)) or using PRO eNSCs (clone 1) following TCF4 and SOX4 RNAi (two RNAi per each targeted gene were combined in plot for TCF4i and SOX4i). Data represent mean \pm SEM (n=3, ANOVA, *P<0.05, **P<0.01). **B** Expression of INSM1 mRNA by qRT-PCR normalized to ACTIN and GAPDH expression in WT eNSCs (clone 1) following INSM1 RNAi. Data represent mean \pm SEM (n=3, ANOVA, **P<0.01). **C** Extreme limiting dilution assay using WT eNSCs (clone 1) following INSM1 RNAi. Data represent mean \pm SEM (n=3, ANOVA, ns). **D** Seahorse XF analysis of glycolysis-dependent extracellular acidification rate in WT eNSCs (clone 1) following INSM1 RNAi. A slight increase in glycolytic activity was observed with only one INSM1 RNAi. Data represent mean \pm SEM. (n=8, ANOVA, ****P<0.0001). **E** INSM1 mRNA expression by qRT-PCR normalized to ACTIN and GAPDH expression in B67 primary human GBM cells following *INSM1* RNAi. Data represent mean \pm SEM (n=3, ANOVA, **P<0.01).

6. Fig. 5: I couldn't find any data related to 'metabolism' in Fig. 5, even though the title is "INSM1 inhibition in PRO eNSCs induces metabolic and developmental regression." Consider putting Extended Data Fig. 7E to Fig. 5 to improve the argument."

As suggested by the reviewer, we have moved previous Supplementary Fig. 7E to main Fig 5F. In addition, we have performed new metabolic experiments using both PRO eNSCs (new Fig. 5I, left) and the primary GBM stem cell line B67 (new Fig. 5I, right), which together demonstrate a reduction in glycolytic capacity following *INSM1* knockdown.

Fig. 5I: Seahorse XF analysis of glycolysis-dependent extracellular acidification rate in in PRO eNSCs (left) and B67 primary human GBM cells (right) following inhibition of INSM1 using two independent RNAi. Data represent mean \pm SEM (GBM: $n=8$; PRO eNSC: $n=4$, ANOVA, **** $P<0.0001$).

7. Fig. 6G: Cluster 6 and Cluster 13 also have a high score of INSM1 signature. What pathways and genes are involved in these clusters? Is there any discussion about them?

This is a good suggestion. We have performed new pathway analyses using differentially expressed gene sets from Clusters 6 and 13 in a new *Supplementary Fig. 11F*.

Clusters 6 and 13 also displayed a relatively high INSM1 signature score and were found to have hybrid cell functions by gene ontology, which included neuronal functions and metabolic responses to cellular stress (*Supplementary Fig. 12B*). We are presenting representative cluster 13.

F**PRO hGBM cluster 13 Gene Ontology**
Supplementary Fig. 11: The INSM1 program in PRO GBM reprograms intermediate neuronal progenitor networks. F Gene ontology analysis of cluster 13-specific genes (n=199). Circle size represents number of genes within each ontology and color indicates log-transformed FDR.

8. Fig. 5H: It should be *INSM1i* on the right side of the y-axis label.

We have corrected this.

9. Line 648: Do you mean *PRO hGBM*?

This now reads PRO human GBM.

10. Line 402: It should be Fig. 6F

We have corrected this.

11. Chromatin immunoprecipitation in the method section: Did you miss describing the conjugation of antibodies to the beads prior to overnight incubation?

We apologize for this omission. We preincubated the ChIP-grade antibodies with magnetic beads by rotation for 30 min at 20-25°C and then isolated antibody-bound beads by magnet before adding sonicated lysates. This is now in the Methods section.

12. *Extended Data Fig. 8: The last x-axis label is off.*

We have corrected this.

13. *Duplicated figures after line 1254.*

We have corrected this.

Reviewer #3

In this manuscript, the authors generated a new in vitro model of glioblastoma by sequentially engineering TERT promoter mutation, p53 loss-of-function mutation, and overexpressing PDGFRA in human neural stem cells. Then, the authors tested the tumorigenicity of these lines in vivo and uncovered that only the triple mutant was able to generate tumors that recapitulate the human glioblastoma. To identify drivers of tumorigenicity, the authors performed a series of global gene expression analyses including CUT/TAG, single-nuclei RNA-seq alone or in couple with ATAC-seq in cell culture model and in vivo tumor xenograft model. These results uncovered that an INSM1-mediated neuronal progenitor network is required for tumor cell evolution. Overall, the manuscript was well written, and the results are well-prepared and presented. The findings should be of interest to many in different fields. As specified below, there are some minor issues that need to be addressed.

1. The addition of the rather weak TERT promoter/p53 LOI data distracts from the main interesting finding that a neuronal progenitor network confers tumorigenicity, and I feel that if the authors had focused on the latter, the paper would have been more compelling.

2. Figure 1C: the wording is confusing. The authors compared OCT/PAX6 expression in ESC to NSC, but not WT to Mutants. In addition, why was PRO not included in the panel?

3. The colors used in the figures are not very visible, especially the yellow color. It is barely visible on paper.

4. Figure 2F&G: WT should be included.

5. Figure 2F: how about the protein levels of TERT?

6. Figure 2H: protein levels of p53 and TERT need to be included.

7. Analysis of scRNA-seq data presented in Figure 3: CytoTRACE was designed for developmental biology. Although cancer cells hijack cell differentiation machinery, the nature of this “differentiation” is fundamentally different from that of developmental biology. If the authors reanalyze their scRNA-seq dataset, e.g., pooling an equal number of cells from each sample and clustering population based on similarity of gene expression, will they come to the same conclusion?

8. Figure 3C: the authors could measure the pH of the media to confirm their Seahorse results.

9. Figure 5: the data presented in Figure 4H revealed that the enrichment of NEU is the only common feature shown in both in vitro and in vivo models. The MTC appeared only in the in vitro model, indicating it might be a cell culture artifact. Therefore, performing experiments shown in Figure 5 in vivo will be more relevant and generate valuable data given that tumorigenicity cannot be modeled only in a cell culture system.

Overall, the manuscript was well-written, and the results are well-prepared and presented. The findings should be of interest to many in different fields. As specified below, there are some minor issues that need to be addressed.

We thank the reviewer for the overall positive comments and have addressed all minor issues raised.

1. The addition of the rather weak TERT promoter/p53 LOI data distracts from the main interesting finding that a neuronal progenitor network confers tumorigenicity, and I feel that if the authors had focused on the later, the paper would have been more compelling.”

As in our response to Reviewer #2, we agree that the finding of increased *TERT* expression with *TP53* loss of function in the context of the *TERT* promoter mutant, however interesting, is not the main focus of the study. We have therefore moved Fig. 2G-I to Supplemental Fig. 3SE-F and have shortened the text in Results regarding this observation.

2. *Figure 1C: the wording is confusing. The authors compared OCT/PAX6 expression in ESC to NSC, but not WT to Mutants. In addition, why was PRO not included in the panel?"*

Fig. 1C shows not only the successful generation of our engineered neural stem cells for each initial genotype but also no significant differences between the expression of cell state markers between each genotype. We originally showed at the mRNA level that wildtype and mutant lines do not differ substantially in terms of cell state markers in ESCs or NSCs (Supplementary Fig. 1D,E). We have now added *new quantification of protein expression of cell state markers by immunofluorescence* in wildtype and mutant lines and find no substantial differences between wildtype and mutant ESCs and NSCs (*new Supplementary Fig. 1C*). We have more explicitly stated in Results that this indicates a negligible effect of genetic background on differentiation status for the purposes of our eNSC GBM model (Lines 103-106). In terms of the PRO eNSCs as mentioned, we have added *new qPCR data for NSC marker PAX6* to demonstrate that expression of this key cell state marker is not altered by addition of PDGFRA mutant to double mutant NSCs (*Supplementary Fig. 1E, right-most graph*).

Supplementary Fig. 1: Engineered NSCs display mutant-specific signaling and malignant transformation in vitro. **A** Table of next generation sequencing (NGS) read counts at *TERT* promoter and TP53 exon 6 CRISPR-mutated sites in two clones (C1 and C2) of wildtype (WT), *TERT*p, and *TERT*p+TP53 hESCs. **B** Immunofluorescence for stem cell markers OCT4 and PAX6 (488 nm), as well as DAPI, in hESC clone set 2 and their differentiation-matched NSCs (scale bar = 50 μ M). **C** Quantification of immunofluorescence intensity normalized to cell number (n=3, ANOVA, ns). **D** Quantitative RT-PCR for analysis of *OCT4* mRNA expression normalized to *ACTIN* and *GAPDH* expression in two independent genetic clone sets of mutant hESC and matched NSC clones (n=3, ANOVA, ***P<0.001, ****P<0.0001). **E** Quantitative RT-PCR for analysis of *PAX6* mRNA expression normalized to *ACTIN* and *GAPDH* expression in two clone sets of hESC and matched NSC clones (n=3, except for right-most histogram where n=2 per clone, ANOVA, *P<0.05, **P<0.01, ***P<0.001). **F** Immunoblot analysis using lysates from PRO eNSCs with indicated antibodies. Representative of three independent replicates. **G** Flow cytometric analysis of EdU incorporation during eNSC in vitro culture (n=5, ANOVA, ns). **H** Extreme limiting dilution assay of serial mutant eNSC genotypes. Data represent mean \pm SEM (n=3, ANOVA, **P<0.01, ****P<0.0001). **I** Representative soft agar colony formation assay of eNSCs (left) and quantification (right). Data represent mean \pm SEM (n=3, ANOVA, **P<0.01, ***P<0.001, ****P<0.0001).

3. The colors used in the figures are not very visible, especially the yellow color. It is barely visible on paper.”

We initially chose the color palette for colorblind accessibility but agree that the yellow color lacks contrast with a white background and obscures visibility. We have chosen a darker shade of yellow and have also made all histogram bars a solid color.

4. Figure 2F&G: WT should be included.

Fig. 2A and B (as well as Supplementary Fig. 3SA) already demonstrate the wildtype to *TERT* promoter mutant comparison for *TERT* mRNA and telomerase activity, both of which are increased in *TERT* promoter mutant eNSCs.

5. Figure 2F: how about the protein levels of TERT?

TERT protein levels are not typically detectable by any of the available commercial antibodies due to its extremely low abundance. Thus in the literature, telomerase activity via telomeric repeat amplification protocol (TRAP) assay (Fig. 2B,G, Supplementary Fig. 3A,C) is commonly shown to represent TERT protein.

Please refer to:

Chiba K, Lorbeer FK, Shain AH, McSwiggen DT, Schruf E, Oh A, Ryu J, Darzacq X, Bastian BC, Hockemeyer D. Mutations in the promoter of the telomerase gene TERT contribute to tumorigenesis by a two-step mechanism. *Science*. 2017 Sep 29;357(6358):1416-1420. doi: 10.1126/science.aao0535. Epub 2017 Aug 17. PMID: 28818973; PMCID: PMC5942222.

Hasegawa K, Zhao Y, Garbuzov A, Corces MR, Neuhöfer P, Gillespie VM, Cheung P, Belk JA, Huang YH, Wei Y, Chen L, Chang HY, Artandi SE. Clonal inactivation of TERT impairs stem cell competition. *Nature*. 2024 Aug;632(8023):201-208. doi: 10.1038/s41586-024-07700-w. Epub 2024 Jul 17. PMID: 39020172; PMCID: PMC11291281.

6. Figure 2H: protein levels of p53 and TERT need to be included.

We have performed new immunoblotting experiments to show *TP53* knockdown at the protein level (see below). Of note, TP53 protein is reliably detectable only in double and triple mutant eNSCs by immunoblotting due to negative feedback by the wildtype TP53, which transactivates MDM2 to target TP53 for degradation (Fig. 1D, Supplementary Fig. 1F) (Haupt et al, *Nature*, 1997). Thus, we are highlighting successful knockdown of TP53 at the protein level only in double mutant *TERT^p+TP53* eNSCs below. The mRNA data for TP53 following RNAi can be found in Supplementary Fig. 3D,E. If the reviewer would like us to include the TP53 immunoblot in the Supplement, we are happy to do this. As mentioned above regarding TERT, protein levels are not typically detectable by immunoblotting.

Immunoblotting for TP53 in *TERT^p+TP53* eNSCs following RNAi.

7. Analysis of scRNA-seq data presented in Figure 3: CytoTRACE was designed for developmental biology. Although cancer cells hijack cell differentiation machinery, the nature of this 'differentiation' is fundamentally different from that of developmental biology. If the authors reanalyze their scRNA-seq

dataset, e.g., pooling an equal number of cells from each sample and clustering population based on similarity of gene expression, will they come to the same conclusion?”

The scRNA data presented in Fig. 3: CytoTRACE was performed using roughly similar cell numbers between experimental groups (WT: 3390 cells, *TERT*^p: 2598 cells, *TERT*^p+*TP53*: 2492 cells, PRO: 2924 cells), thus we would expect downsampling to yield similar results in terms of differentiation order. We would also point out that the RNA velocity analysis orthogonally validates our CytoTRACE findings (Fig. 3D,E)

8. Figure 3C: the authors could measure the pH of the media to confirm their Seahorse results.

The extracellular acidification rate calculated during Seahorse XF analysis is based on repeat measurements of free proton concentration—essentially media pH—using micro-well technology as a proxy for glycolysis. Per the company website (<https://www.agilent.com/en/products/cell-analysis/how-seahorse-xf-analyzers-work>):

Solid-state sensor probes create a transient microchamber 200 microns above the cell monolayer within a microplate. They then measure the changes in concentrations of dissolved oxygen and free protons, caused by cellular oxygen consumption (respiration) and proton excretion (glycolysis). Following these measurements for 2–5 minutes, the instrument calculates the OCR and ECAR, respectively.

9. Figure 5: the data presented in Figure 4H revealed that the enrichment of NEU is the only common feature shown in both *in vitro* and *in vivo* models. The MTC appeared only in the *in vitro* model, indicating it might be a cell culture artifact. Therefore, performing experiments shown in Figure 5 *in vivo* will be more relevant and generate valuable data given that tumorigenicity cannot be modeled only in a cell culture system.

We agree that tumorigenicity cannot be perfectly modeled *in vitro* and have therefore added *new in vivo* data to test the role of INSM1 in PRO eNSC tumorigenicity *in vivo* (new Fig. 6E,F). We orthotopically transplanted *INSM1* knockdown PRO eNSCs using two independent RNAi hairpins (vs. control RNAi) into the brains of athymic nude mice. Importantly, mice bearing *INSM1* knockdown PRO eNSCs exhibited increased survival compared to mice bearing control RNAi-infected PRO eNSCs (Fig. 6E). *INSM1* RNAi inhibited brain tumor formation by PRO eNSCs as monitored by MRI (Fig. 6F). These results indicate that *INSM1* is critical for the *in vivo* tumorigenicity of PRO cells. We agree that the ideal experimental design for transcriptional analysis of *INSM1* knockdown PRO eNSCs would be in the *in vivo* setting. However, harvesting an adequate number of cells from *INSM1* knockdown PRO eNSC-derived tumors from mouse brain posed a technical challenge since tumors from *INSM1* knockdown PRO tumors form no or very small tumors (please see MRIs in Fig. 6F).

Fig. 6: INSM1 is upregulated in human GBM tumors and is critical for *in vivo* tumorigenicity. E Kaplan-Meier survival analysis of PRO eNSCs (2 clones combined) with INSM1 inhibited by two

independent RNAi. Cells were orthotopically implanted in mice (clone 1: $n=10$; clone 2: $n=5$, log-rank, **** $P<0.0001$). F T2-weighted MRI images of mice 200 days post-xenograft with first clonal set of four serial mutant eNSCs. Red outlines indicate pathological lesions in the coronal plane of the injection site (scale bar = 1 mm; L = left, P = posterior, R = right).

Reviewer #4

In the submitted manuscript, the authors provide a human framework to introduce oncogenic drivers and mutations in specific neural cell lineages in a step-wise fashion. This is accompanied by detailed transcriptional and epigenetic characterization of tumor-associated cell states tied to specific and combinations of oncogenic drivers. In glioblastoma, this has relevance given the cellular diversity of transcriptionally distinct populations, and their significant role in disease progression and therapeutic resistance. From a functional perspective, the authors demonstrate the disruption of key factors that regulate proneural cell state, specifically INSM1 inhibition, have important effects on cancer stem cell frequency and transcriptional lineage programs, that may have functional impact on GBM tumor development. While the isogenic modeling of GBM and associated cell states is of particular relevance, in addition to studies of transcriptional evolution, several outstanding questions remain to be addressed:

1) It's clear from the author's detailed transcriptional analysis that PRO tumor models have cell types that can be classified as proneural, mesenchymal, and classical. However, unclear is how transcriptionally similar are tumors to primary GBM tumors, or proneural signatures from GBM patient samples. Additionally, would DNA methylation classification of PRO eNSC tumors place these models in similarity with primary GBM tumors, and those enriched with proneural signatures.

2) The functional impact on INSM1 and stem cell state is not clearly defined. How does INSM1 loss-of-function alter tumor initiation, of the tumors (assuming they arise) now enriched in other GBM cell states that are capable of driving tumor development.

3) The importance of INSM1 and its effects on proneural state need to be validated in patient derived tumor models. A related concern is whether cell lineage programs modeled are reflective of neural stem cell populations seen in the adult setting, where glioblastoma multiforme is more frequently observed.

1) It's clear from the author's detailed transcriptional analysis that PRO tumor models have cell types that can be classified as proneural, mesenchymal, and classical. However, unclear is how transcriptionally similar are tumors to primary GBM tumors, or proneural signatures from GBM patient samples. Additionally, would DNA methylation classification of PRO eNSC tumors place these models in similarity with primary GBM tumors, and those enriched with proneural signatures.”

To address the reviewer's comment, we have performed additional analyses by hierarchical clustering of pseudobulk expression from PRO eNSC brain tumor scRNAseq data with bulk RNA data from The Cancer Genome Atlas (TCGA) GBM patient tumors and observed close proximity of PRO tumors to GBM samples classified as the PRO subtype with distinct separation from those classified as Mesenchymal or Classical (*new Supplementary Fig. 7B*).

B

Supplementary Fig. 7B: Hierarchical clustering of TCGA-GBM patient sample bulk RNA subtypes with PRO eNSC tumor pseudobulk expression data from the scRNAseq dataset. Red asterisk indicates PRO eNSC sample.

2) *The functional impact on INSM1 and stem cell state is not clearly defined. How does INSM1 loss-of-function alter tumor initiation of the tumors (assuming they arise) now enriched in other GBM cell states that are capable of driving tumor development.*

We agree that functional validation of the role of INSM1 in tumor initiation is needed. We have therefore performed *new in vivo* experiments to test the role of INSM1 in PRO eNSC tumorigenicity *in vivo* (new Fig. 6E,F). We orthotopically transplanted *INSM1* knockdown PRO eNSCs using two independent RNAi hairpins (vs. control RNAi) into the brains of athymic nude mice. Importantly, mice bearing *INSM1* knockdown PRO eNSCs exhibited increased survival compared to mice bearing control RNAi-infected PRO eNSCs. *INSM1* RNAi inhibited brain tumor formation by PRO eNSCs as monitored by MRI (Fig. 6F). These results indicate that INSM1 is critical for the *in vivo* tumorigenicity of PRO cells. Given the expression of *INSM1* in the intermediate progenitor cell (IPC) during human cortical development and the presence of an IPC-like cell state both in our PRO eNSC brain tumors and in human GBM tumors (Fig. 7), we hypothesize that INSM1 governs an IPC-like state that is tumorigenic in human GBM. This represents a new concept in GBM tumor pathogenesis, and further studies on the mechanisms of INSM1-driven tumor initiation are warranted in future studies.

Fig. 6: INSM1 is upregulated in human GBM tumors and is critical for *in vivo* tumorigenicity. E Kaplan-Meier survival analysis of PRO eNSCs (2 clones combined) with INSM1 inhibited by two independent RNAi. Cells were orthotopically implanted in mice (clone 1: $n=10$; clone 2: $n=5$, log-rank, **** $P<0.0001$). F T2-weighted MRI images of mice 200 days post-xenograft with first clonal set of four serial mutant eNSCs. Red outlines indicate pathological lesions in the coronal plane of the injection site (scale bar = 1 mm; L = left, P = posterior, R = right).

3) *The importance of INSM1 and its effects on proneural state need to be validated in patient derived tumor models. A related concern is whether cell lineage programs modeled are reflective of neural stem cell populations seen in the adult setting, where glioblastoma multiforme is more frequently observed.*

We agree that orthogonal functional validation using primary GBM cells is needed. Given our experience with patient-derived primary glioblastoma stem cell (GSC) cultures (Mao, Gujar, et al, *Cell Rep*, 2015; Gujar et al, *PNAS*, 2016; Mahlokozera et al, *Nat Comm*, 2021), we have performed *new* metabolic experiments using the B67 GSC line in the setting of *INSM1* knockdown, which indicate that INSM1 is required for glycolytic activity in primary GBM cells as predicted from our PRO eNSC model (new Fig 5I).

B67 primary GBM

Fig. 5: INSM1 inhibition in PRO eNSCs induces metabolic and developmental regression. I Seahorse XF analysis of glycolysis-dependent extracellular acidification rate in primary human GBM cells following inhibition of *INSM1* using two independent RNAi. Data represent mean \pm SEM (GBM: $n=8$, ANOVA, **** $P<0.0001$).

Point-By-Point Response to Reviewers' Comments

Manuscript#: NCOMMS-23-36248B

Title: An aberrant INSM1-dependent intermediate neuronal progenitor state drives tumorigenesis in a human stem cell model of glioblastoma

We appreciate the reviewers' insightful comments and suggestions. Below, we provide our detailed response to each comment and description of related revisions in our manuscript.

Reviewer #2

The authors have addressed all my concerns. The additional pathway analysis and Seahorse XF analysis support the hypothesis that gene expression shifts from promoting metabolism to neurodevelopment during tumor evolution.

We appreciate the reviewer's thoughtful and balanced feedback on our study revisions.

Reviewer #3

I appreciate the effort the authors have made to address the concerns I previously raised. I am pleased to see that most of my concerns have been adequately addressed, including clarifications in figure legends, additional experimental validations, and improvements in data presentation. The inclusion of in vivo experiments to validate the role of INSM1 further strengthens the impact of this study.

However, I still have concerns regarding the scRNA-seq analysis presented in Figure 3. While I understand that the dataset was processed with roughly similar cell numbers across experimental groups and that RNA velocity was used as an orthogonal validation, my concern remains regarding the appropriateness of CytoTRACE for this context. Given the fundamental differences between developmental differentiation and cancer cell state transitions, I would strongly encourage a re-analysis of the scRNA-seq dataset using an alternative clustering approach—such as unbiased clustering based on transcriptomic similarity across pooled cells from all conditions. This would help confirm whether the observed differentiation hierarchy remains consistent across methods. Overall, I find the manuscript significantly improved and believe this additional analysis will further strengthen the conclusions.

We appreciate the reviewer's thoughtful comments on the need for scientific rigor, particularly when using computational approaches. We understand the concerns regarding the appropriateness of applying an algorithm designed for developmental processes to a cancer evolution context. As mentioned in Results, we did attempt standard principal component analysis (PCA) prior to both CytoTRACE and RNA velocity analyses. However, the UMAP reduction constrained by PCA did not meaningfully differentiate among our various mutant genotypes, particularly using pseudotime inference (data not shown), suggesting that PCA could not parse mutant-specific expression changes from background variations in the developmental neural stem cell state. In fact, this was our rationale for moving to developmentally oriented approaches. Interestingly, though, we further investigated the principal components (PC) and correlated genes identified in our PCA and did observe some signals specific to our mutant genotypes and consistent with our other gene expression and chromatin accessibility changes. We observed that up to 20 PCs captured the majority of variation represented in the serial mutation scRNA-seq dataset and that PCs 9-16 appeared to demonstrate specificity for mutant-specific genetic backgrounds, particularly double and triple mutant PRO eNSCs. Interestingly, genes of interest represented in these PCs included SOX4 (PC12), PDGFRA (PC15), and INSM1 (PC16). However, given the limited interpretability of pseudotime inference in our PCA-constrained dimension reduction, determining the directionality of PC-associated vectors and the biological significance of correlated vs. anti-correlated genes remained challenging. Thus, while PCA demonstrated

some signals that support our orthogonal analyses, we believe it remained limited in its utility for discovery in our dataset. We have thus not added these analyses to the manuscript but can do so if requested.

In vitro PRO eNSC serial mutant scRNA PCA. **A** Elbow plot of *in vitro* serial mutant PRO eNSC scRNA dataset demonstrating the variation captured by each principal component (PC). **B** Dot plot showing association of PCs with serial mutant eNSC genotypes. **C** Genes most correlated (PC>0) or anti-correlated (PC<0) with PCs 12, 15, 16. Arrowheads indicate genes of interest identified during previous analyses.

Reviewer #4

In the revised manuscript, the authors have responded with new in vitro, in silico, and in vivo data and addressed my initial comments thoroughly. I have no further or new comments.

We thank the reviewer for these kind comments.

Reviewer #5 (Remarks to the Author)

Reviewer #6-Replacement for Reviewer #1 (Remarks to the Author)

The authors are highly responsive to. The authors adequately addressed most of the comments by reviewer A with new supporting data and satisfactorily responded. However, the response to comment #6 “There is incremental buildup of inference upon probability in the figures beyond figure 2 for which there is no systematic functional validation. Omic analyses must be backed by biological testing and validation” is insufficient.

We would first respectfully point out to the reviewer the multiple lines of functional validation experiments beyond Figure 2 in the first revision, *many of which were new revision experiments* (Functional experiments in prior revision according to current revision figure designation: Figure 3C, Figure 5A,H,I, Figure 6E,F, Supplementary Figure 4F, Supplementary Figure 10A-D). In addition, in this second revision, we now present *new functional experiments* targeting INSM1 in primary glioblastoma stem-like cells (GSCs), further supporting its role in human GBM. INSM1 knockdown in primary human GSC lines significantly reduced stem cell frequency, as demonstrated by limiting dilution self-renewal assays in *new Supplementary Figures 10E-F*. These results underscore INSM1’s importance in maintaining malignant stem-like phenotypes across both engineered and patient-derived GBM models. Additional context for these experiments is provided in our response to Comment #3 below.

Supplementary Fig. 10E: INSM1 mRNA expression by qRT-PCR normalized to ACTIN and GAPDH expression in B165 (left) and B188 (middle) primary GSCs following *INSM1* RNAi. Data represent mean \pm SEM (B165: $n=3$; B188: $n=2$, Student's t-test, $*P<0.05$). **F** Extreme limiting dilution assay in aggregated B165 and B188 primary GSCs following inhibition of *INSM1* using RNAi. Data represent mean \pm SEM (B165: $n=2$; B188: $n=2$, Student's t-test, $*P<0.05$).

New Comments:

1. *PRO (TERT^p C228T, TP53 G473A @ R248Q, PDGFRAD284V) model does not recapitulate the actual PDGFRA mutation in clinical adult GBM tumors. The PDGFRAD284V mutation occurs in pediatric high-grade glioma (pHGG) in children and often associated with H3 K27M mutated pHGG. In adult GBM, PDGFRAD284V mutation is rare or highly infrequent. In contrast, a PDGFRA Δ 7-8 mutation has been reported and characterized in adult GBM tumors. Thus, the PRO eNSC GBM model presented in this study does not recapitulate clinical presentation of PDGFRA mutation in adult GBM.*

The frequent co-occurrence of activating mutations with gene-level amplification events in PDGFRA suggests that large-scale oncogenic mobilization of this receptor tyrosine kinase (RTK) pathway is crucial for GBM pathogenesis (PMID: 20889717; PMID: 23074200; PMID: 35911637). In reference to “*PDGFRA Δ 7-8 mutation*,” we assume that the reviewer is referring to the common in-frame deletion of PDGFRA exons 8 and 9 in glioblastoma (PDGFRA ^{Δ 8-9} mutation), which is observed at a higher frequency in adult human GBM than the PDGFRA^{D842V} mutation. Nevertheless, the PDGFRA^{D842V} mutation has been found to occur at a rate of 1-2% in adult human GBM in a Chinese cohort (doi.org/10.1016/j.annonc.2023.09.1713) and examples of adult GBMs with the PDGFRA^{D842V} mutation have been reported (cbioportal.org). Due to our desire to utilize a kinase-active PDGFRA mutation and the fact that the PDGFRA^{D842V} mutation does occur in adult GBM, we chose to model PRO GBMs with this PDGFRA mutation. Introduction of this PDGFRA-activating point mutation to our model resulted in dramatic PDGFRA protein kinase activation by immunoblotting (Figure 1D) along with activation of downstream signaling effectors like AKT and ERK. Most importantly, introduction of the PDGFRA^{D842V} mutant to *TERT* promoter and *TP53* double mutant eNSCs conferred robust malignant transformation with *in vivo* tumor formation (Figure 1H-K). Thus, we believe the addition of PDGFRA^{D842V} mutation in our system appropriately models the functional consequences of perturbing this RTK pathway, which is often observed in adult human GBM. We view the use of the PDGFRA ^{Δ 8-9} mutation in our model system as an opportunity for future studies, where we could compare the malignant potential of different RTK mutants.

2. *The advance of development and characterization of the PRO eNSC GBM model system in relation to recently reported human induced pluripotent stem cell (hiPSC)-derived glioma avatar models is not clear.*

The rationale of using human embryonic stem cells versus neural stem cell as the starting cell model is not strong. The response by the authors to this comment is insufficient.

We employed NIH-approved human embryonic stem cell (hESC) lines to generate our GBM model, as these lines offer a highly reproducible and standardized platform. Their ready availability and rigorous quality control by authorized distributors ensure experimental consistency across laboratories and studies. While we selected hESCs for these advantages, we strongly support the use of both hESCs and human induced pluripotent stem cells (iPSCs) as valuable and complementary tools for modeling tumor biology in oncology research.

3. The rationale of selection of INSM1 as the key transcription factor downstream of PRO mutation drivers is weak. As shown in Figure 6D, in relation to SOX4 and TCF4, the actual level of INSM1 in human GBM is much lower. The authors should also present the relative expression of other TFs in 6D, STMN2, DCX, SOX4, TCF4 and CD24 in the form of 6B and justify the focus of INSM1 in this study.

We respectfully disagree with the reviewer and would like to clarify the rationale for prioritizing INSM1 as a transcription factor central to malignant phenotypes in Proneural (PRO) GBM. A growing body of evidence from our study supports this selection:

We first identified INSM1 through an unbiased chromatin and expression approach using integrated scRNA-seq and scATAC-seq (Figure 3H), which nominated INSM1 as a candidate transcription factor associated with transformation. This was followed by evidence of an association between INSM1 gene expression and late evolution during *in vivo* tumorigenesis (Figure 4D–E), highlighting its potential relevance during malignant progression. Our initial functional evidence supporting a role for INSM1 in PRO eNSCs was that its knockdown induced broad and profound transcriptomic changes towards a wildtype-like state (Figures 5B–C, 5F). This effect was specific to INSM1; knockdown of other transcription factors with higher relative expression in GBM, such as TCF4 and SOX4, did not recapitulate this reversion. These findings argue that expression level alone is not a reliable indicator of functional importance.

As an illustrative example, TERT mRNA may be expressed at low levels in cancer cells (e.g., ~20 molecules per cell) yet produce substantial protein output (~250 copies) essential for overcoming cellular senescence (PMID: 17395830; PMID: 24990373; PMID: 11726691). This underscores that low mRNA abundance does not preclude functional significance, especially for factors like INSM1 that may require precise temporal control during key developmental windows. INSM1 is known to act as a transcriptional repressor during neural development, and its tight regulation may reflect similar context-dependent control in GBM, although this remains to be validated. Moreover, protein levels of transcription factors can frequently diverge from mRNA levels due to post-transcriptional regulation—an important consideration beyond the current study's scope.

Finally, as mentioned above, in addition to the multiple functional experiments we have provided in the setting of INSM1 inhibition in PRO eNSCs and primary glioblastoma stem-like cells (GSC) (Figure 5H–I, Figure 6E–F), we have now added *new functional evidence* further supporting INSM1's role in human GBM. In primary human GSC lines, INSM1 knockdown significantly reduced stem cell frequency, as assessed by limiting dilution self-renewal assays in a *new Supplementary Figure 10E–F*. These findings reinforce INSM1's critical role in maintaining malignant stem-like phenotypes in both engineered and patient-derived GBM models.

Supplementary Fig. 10E: INSM1 mRNA expression by qRT-PCR normalized to ACTIN and GAPDH expression in B165 (left) and B188 (middle) primary GSCs following *INSM1* RNAi. Data represent mean \pm SEM (B165: $n=3$; B188: $n=2$, Student's t-test, $*P<0.05$). **F** Extreme limiting dilution assay in aggregated B165 and B188 primary GSCs following inhibition of *INSM1* using RNAi. Data represent mean \pm SEM (B165: $n=2$; B188: $n=2$, Student's t-test, $*P<0.05$).

4. The cellular function of *INSM1* is transcriptional repressor in neurogenesis and neuroendocrine cell differentiation during embryonic and/or fetal development. *INSM1* represses gene transcription by recruiting chromatin-modifying factors, such as HDAC1, HDAC2, KDM1, and RCOR1 histone deacetylases. However, the functional and consequential studies by knockdown of *INSM1* (*INSMi*) in PRO models presented here failed to relate to the basic cellular function of *INSM1*.

We thank the reviewer for this comment and agree that the molecular functions of *INSM1* as a transcriptional repressor via recruitment of chromatin-modifying factors during development is of great interest to us in the GBM context. However, given the large amount of data currently presented in the manuscript and the already complex narrative, we believe that determining the precise molecular mechanisms of *INSM1* in GBM is outside the scope of the current study. We plan to intensely investigate this question and hope to share exciting, new findings in a future publication.

Point-By-Point Response to Reviewers' Comments

Manuscript#: NCOMMS-23-36248C

Title: INSM1 governs a neuronal progenitor state that drives glioblastoma in a human stem cell model

Reviewer #3

I appreciate the authors' efforts to address my concerns regarding the scRNA-seq analysis. The inclusion of RNA velocity as an orthogonal validation is a valuable addition, and the roughly similar cell numbers across experimental groups partially mitigate the sampling bias concerns. However, a few issues remain:

CytoTRACE Suitability: While RNA velocity supports the trajectory findings, the fundamental concern about CytoTRACE's applicability to cancer contexts—where "differentiation" differs from developmental processes—remains unaddressed. The authors should acknowledge this limitation or test an alternative trajectory method (e.g., Monocle 3 or Slingshot) to confirm their results.

Alternative Analysis: The authors did not perform the suggested downsampling to equal cell numbers and re-clustering, which would have directly tested the robustness of their conclusions. Although their existing pooled clustering approach is reasonable, this exact reanalysis would strengthen confidence in the results.

Given the overall strength of the study and the authors' good-faith response, I recommend minor revisions to address these points. Specifically, the authors should:

Discuss the limitations of applying CytoTRACE to cancer data.

Consider performing the downsampling and re-clustering analysis or provide a stronger justification for why it is unnecessary.

These revisions will enhance the manuscript's rigor without requiring extensive new experiments.

We would like to thank the reviewer for their continued careful consideration of our manuscript. In the text, we now discuss both these remaining points, saying:

“We utilized Monocle3 for graph-based pseudotime analysis to validate the biological significance of inferred ordering in the dimension-reduced splicing space (Supplementary Fig. 5A), and observed concordant temporal trajectories terminating in the mutant-specific transcriptional landscape.”

This sentence has been added in the Results section for Fig. 3D-E, and the use of Monocle3 has been more explicitly stated and cited in the Methods section. This point is further discussed in the Discussion section as follows:

“Leveraging joint multiomic profiling of our eNSCs in vitro enabled high-resolution network mapping of TFs controlling fate specification in mutant eNSCs, which we validated through orthogonal splicing-based lineage trajectory analyses of PRO eNSC tumors that formed in vivo. We identified chromatin regulators that consistently appeared as significant expression changes during transcriptome evolution (Supplementary Fig. 6E-G), including SOX4 and TCF4, which have been implicated in invasive signatures of GBM. Of particular interest, INSM1 has been demonstrated to play a defining role in the identity of IPCs during development and their delamination from the subventricular zone to generate outer cortical neurons. While mounting evidence suggests that normal neurodevelopment mechanisms are hijacked by GBM cells, it is important to note that trajectory analysis based on scRNA-seq data (e.g., CytoTRACE and Dynamo) relies on the assumption that cancer progression obeys principles of physiologic RNA dynamics. Alternative methods (e.g., Monocle3 or Slingshot) infer trajectory from models constructed following dimensional reduction, as opposed to biological metrics, such as number of genes expressed or RNA splicing, utilized

by CytoTRACE and Dynamo, respectively. Though preliminary analyses with these methods validate the biological significance of modelling RNA dynamics in our data (Supplementary Fig. 5A), future studies analyzing these datasets using alternative analyses could provide new insights and reduce certain technical or biological variations. For example, downsampling in scRNA-seq can be useful for standardization but inherently discards data, which can reduce sensitivity for detecting rare populations, obscure subtle biological differences, and introduce stochastic variation due to the random removal of reads or cells. We chose not to downsample in this study to avoid these issues. Nevertheless, we leveraged our transcriptomic analyses in eNSCs to define an oncogenic INSM1 signature that exhibited upregulation in later cycling subpopulations during evolution of a human PRO GBM tumor.”